# Towards hybrid modeling of the global hydrological cycle

Basil Kraft[1,2], Martin Jung[1], Marco Körner[2], Sujan Koirala[1], and Markus Reichstein[1]

[1]Department of Biogeochemical Integration, Max Planck Institute for Biogeochemistry, Germany
[2]Department of Aerospace and Geodesy, Technical University of Munich, Germany

**Correspondence:** Basil Kraft (bkraft@bgc-jena.mpg.de)

**Abstract.**

State-of-the-art global hydrological models (GHMs) exhibit large uncertainties in hydrological simulations due to the complexity, diversity, and heterogeneity of the land surface and subsurface processes, as well as scale-dependency of these processes and associated parameters. Recent progress in machine learning, fueled by relevant Earth observation data streams, may help overcome these challenges. But machine learning methods are not bound by physical laws and their interpretability is limited by design.

In this study, we exemplify a hybrid approach to global hydrological modeling that exploits the data-adaptivity of neural networks for representing uncertain processes within a model structure based on physical principles (e.g., mass conservation), that form the basis of GHMs. This combination of machine learning and physical knowledge can potentially lead to data-driven, yet physically consistent and partially interpretable hybrid models.

The hybrid hydrological model (H2M), extended from Kraft et al. (2020), simulates the dynamics of snow, soil moisture, and groundwater storage globally at 1° spatial resolution and daily time step. Water fluxes are simulated by an embedded recurrent neural network. We trained the model simultaneously against observational products of terrestrial water storage variations (TWS), grid cell runoff (Q), evapotranspiration (ET), and snow water equivalent (SWE) with a multi-task learning approach.

We find that the H2M is capable of reproducing key patterns of global water cycle components with model performances being at least on par with four state-of-the-art GHMs, which provide a necessary benchmark for H2M. The neural network learned hydrological responses of evapotranspiration and grid cell runoff to antecedent soil moisture states qualitatively consistent with our understanding and theory. The simulated contributions of groundwater, soil moisture, and snowpack variability to TWS variations are plausible and within the ranges of traditional GHMs. H2M identifies a somewhat stronger role of soil moisture for TWS variations in transitional and tropical regions compared to GHMs.

With the findings and analysis, we conclude that H2M provides a new data-driven perspective on modeling the global hydrological cycle and physical responses with machine-learned parameters, that is consistent with and complementary to existing global modeling frameworks. The hybrid modeling approaches have a large potential to better leverage ever-increasing Earth observation data streams to advance our understandings of the Earth system and capabilities to monitor and model it.

## 1 Introduction

Physically-based global hydrological models (GHMs) are an essential tool to understand, monitor, and forecast the water cycle with an array of societal implications (Jiménez Cisneros et al., 2014). Yet, GHMs and land-surface models face many challenges related to process representations and parameterizations, resulting in large uncertainties (Schellekens et al., 2017). The existing state-of-the-art GHMs still disagree across all spatial and temporal scales, which may be attributed to limited, biased, and uncertain data, the heterogeneity of considered processes, or a lack of process understanding (Haddeland et al., 2011; Beck et al., 2017). While global water cycle observations are increasing rapidly, a thorough integration with a GHM to overcome uncertainties is rarely facilitated due to the model complexity and computational expenses, even though some GHMs use data, e.g., river discharge, to calibrate model parameters (e.g., Van Beek et al., 2011).

Different pathways have been proposed to utilize additional Earth observation data in hydrological modeling. For instance, physically-based models benefit from using spatially explicit parameters, which can be retrieved from Earth observation data. It is, for example, common to use spatio-temporally varying leaf area index as a model parameter (e.g., Van Der Knijff et al., 2010) to account for vegetation dynamics. Furthermore, upscaling of locally-estimated or measured parameters to global scale—such as catchment parameters (Beck et al., 2016) or soil properties (Hengl et al., 2017)—can improve model accuracy. Using model-data-integration approaches, it has been shown that relatively simple conceptual hydrological models can yield state-of-the-art performance when calibrated simultaneously on multiple observational data constraints (Trautmann et al., 2018), which opens new avenues for targeted, partially data-driven experiments to parameterize hydrological processes.

Other approaches to integrate additional observations and physically-based models have been developed in the domain of data assimilation (McLaughlin, 2002; Reichle, 2008). While classic data assimilation aims to correct model states or provide initial conditions using additional observational data (Sun et al., 2016), promising concepts exist to learn time-varying model parameters from data (Moradkhani et al., 2005; Geer, 2021). If system understanding and out-of-sample performance (e.g., long-term prediction) are not central, the use of (purely data-driven) deep learning approaches has been proposed and applied recently in hydrology, and experimental methods for gaining (so far only qualitative) insights exist (Shen et al., 2018).

Recently, it has been proposed to fuse process models with machine learning into one end-to-end modeling system, in the so-called hybrid modeling approaches (Reichstein et al., 2019). The hybrid approaches aim at harvesting the information in Earth observation data efficiently by replacing uncertain parameters and processes with a machine learning model, while still maintaining model interpretability and physical consistency. Furthermore, the approach facilitates the incorporation and integration of information from multiple data sources, which is a bottleneck in GHMs. Hybrid modeling can be employed to improve the predictability of the Earth system or components thereof, such as sea surface temperature (de Bézenac et al., 2019), or subgrid atmospheric processes (Rasp et al., 2018). Alternatively but not mutually exclusive, hybrid modeling can leverage the flexibility of machine learning models with the goal to retrieve data-driven, yet interpretable physical coefficients and latent variables.

One of the key hydrological data products for diagnosing and understanding global land water cycle variations is total terrestrial water storage (TWS). The TWS is an observation-based rasterized product that integrates all water storage components

and is used for calibration and validation of process-based models (Güntner et al., 2007; Schellekens et al., 2017; Trautmann et al., 2018; Scanlon et al., 2019) and in data-driven studies (Humphrey et al., 2016; Andrew et al., 2017; Rodell et al., 2018). An attribution of TWS variations to its components is still unclear as current model simulations do not produce consistent spatio-temporal patterns due to uncertainties in the model structure and process description, forcing data, and parameter values (Güntner, 2008). Such attribution is not trivial, especially as contiguous observations of the storage components are not available separately on a global scale (e.g., groundwater) or limited (e.g., soil moisture, where satellite observations are only representative of the top soil layers). Thus, decomposition of TWS components is either done with large-scale hydrological modeling (Schellekens et al., 2017), locally using in-situ data (e.g., Swenson et al., 2008), or with data-driven approaches without a strict constraint on physical consistency (Andrew et al., 2017).

This study aims to complement and bridge the previous global-scale hydrological modeling and observation-based syntheses by comprehensively evaluating the potential of hybrid modeling at the global scale. In particular, it provides a much-needed data-driven perspective on the global water cycle and its spatio-temporal variability based on carefully designed cross-validation analysis, and that with a crucial consideration of the basic physical principle of mass conservation. To do so, we have further developed the model proposed by Kraft et al. (2020), especially with regards to model robustness and physical consistency. The overarching goal of this study is to provide a comprehensive description and assessment of the applicability of the hybrid modeling approach as a potential novel avenue for global hydrological simulation. Particular emphasis are put on benchmarking against and complementing state-of-the-art hydrological models and assessing the plausibility and interpretability of the machine learning–based data-driven hydrological responses going beyond typical focus on predictive skills. Furthermore, we examine the potential applications and limitations on a challenging use case of decomposing the contributions of different water storage components to the variations of TWS.

We first describe the datasets used, the hybrid hydrological model (H2M), and the model training and evaluation approach in Section 2. We then show the H2M performance in Sect. 3.1 and present the benchmarking against a set of GHM simulations from the eartH2Observe ensemble in Sect. 3.2. Section 3.3 provides the data-driven perspective on hydrological responses, followed by Sect. 3.4 that focuses on the TWS decomposition. Additional plausibility and interpretability of the H2M simulations are presented in Sect. 4.1 and Sect. 4.2. Lastly, we provide a more general assessment of the challenges and opportunities of the hybrid approach in Sect. 4.3.

## 2 Data and methods

### 2.1 Datasets

#### 2.1.1 Meteorological forcing

Three time-varying meteorological datasets were used to force H2M (Tab. 1):

**i)** Precipitation observations, obtained from the Global Precipitation Climatology Project dataset (GPCP-1DD) v1.2 (Huffman et al., 2012).

**Table 1.** Dataset overview: water cycle constraints, meteorological forcing and static variables with their native and aggregated spatial, as well as their temporal resolution. We use upper case for state variables and lower case for fluxes in the mathematical notation.

| | Acr. | Math. notation | Spatial resolution | | Temporal resolution | Dataset | Resources |
|---|---|---|---|---|---|---|---|
| | | | Native | Agg. | | | |
| **Water cycle constraints** | | | | | | | |
| Terrestrial water storage | TWS | $T$ | 0.50° | 1.00° | Monthly | GRACE Tellus JPL RL06M v1 | Watkins et al. (2015), Wiese et al. (2018) |
| Evapotranspiration | ET | $e$ | 0.50° | 1.00° | Monthly | FLUXCOM v1 | Tramontana et al. (2016), Jung et al. (2019) |
| Grid cell runoff | Q | $q$ | 0.50° | 1.00° | Monthly | GRUN v1 | Ghiggi et al. (2019) |
| Snow water equivalent | SWE | $S$ | 0.25° | 1.00° | Daily | GlobSnow v2 | Takala et al. (2011), Luojus et al. (2014) |
| **Meteorological forcing** | | | | | | | |
| Precipitation | - | $p$ | 1.00° | 1.00° | Daily | GPCP 1dd v1.2 | Huffman et al. (2012) |
| Net radiation | - | $r_{\mathrm{net}}$ | 1.00° | 1.00° | Daily | CERES SYN1deg Ed4A | Wielicki et al. (1996), Doelling (2017) |
| Air temperature | - | $T_{\mathrm{air}}$ | 0.50° | 1.00° | Daily | CRUNCEP v8 | Harris et al. (2014), Viovy (2018) |
| **Static variables** | | | | | | | |
| Soil properties | - | - | 1/120° | 1/30° | - | Soilgrids v2 | Hengl et al. (2017) |
| Land cover fractions | - | - | 1/360° | 1/30° | - | Globland30 v1 | Chen et al. (2015) |
| Digital elevation model | - | - | 1/120° | 1/30° | - | GTOPO | DOI/USGS/EROS (1997) |
| Wetlands | - | - | 1/240° | 1/30° | - | Tootchi | Tootchi et al. (2019) |

Arc.=acronym, Agg.=aggregated

**ii)** Net radiation, provided by the SYN1deg Ed3A product (Doelling, 2017) of the Clouds and the Earth's Radiant Energy Systems (CERES) program (Wielicki et al., 1996).

**iii)** Air temperature, obtained from CRUNCEP v8 dataset, a product of the observation-based Climate Research Unit (CRU) and the National Center for Environmental Prediction (NCEP) reanalysis data (Harris et al., 2014; Viovy, 2018).

To test the impact of the model forcings on the comparison with GHMs (Sect. 3.2), we carried out additional H2M simulation with forcing datasets from the Watch Forcing Data-ERA Interim (WFDEI) dataset (Weedon et al., 2014) in an independent setup (Appendix D).

### 2.1.2 Static variables

A set of temporally static variables was used to represent land surface characteristics (Tab. 1):

**i)** Soil properties from the soilgrids dataset (Hengl et al., 2017): *absolute depth to bedrock* and the average (along depth) of *bulk density*, *coarse fragments*, *clay*, *silt*, and *sand* (6 variables in total).

**ii)** Land cover fractions from the Globland30 dataset (Chen et al., 2015) for the 10 classes: *water bodies*, *wetlands*, *artificial surfaces*, *tundra*, *permanent snow and ice*, *grasslands*, *barren*, *cultivated land*, *shrublands*, and *forests*.

**iii)** Digital elevation model from GTOPO30 (DOI/USGS/EROS, 1997).

    **iv)** Fractions of groundwater-driven wetlands, regularly flooded wetlands, and the intersection of them (Tootchi et al., 2019), i.e., a total of 3 variables.

    These 20 static variables were spatially aggregated from their finer resolution to $1/30°$ to maintain sub-grid variations, yielding a block of 30 latitude cells times 30 longitude cells times 20 variables, i.e., a total of 18 000 values per $1°$ grid cell,

the spatial resolution of the forcing data. Due to the high dimensionality of the static variables, the data was compressed in a pre-processing step using a simple convolutional autoencoder, consisting of an encoder, a bottleneck layer, and a decoder. The decoder is a stack of consecutively smaller convolutional neural network (CNN) layers that reduce the input block to a vector of size 30, the bottleneck layer. This process is then reverted in the decoder model, mapping the vector back to the input data. The CNN model is optimized to reconstruct the input data but is forced to find a low-dimensional representation enforced by

the bottleneck (e.g., Goodfellow et al., 2016). The resulting compressed dataset consists of 30 latent variables per grid cell that encode the original high-dimensional data (18 000), which is then used as an input to H2M (Section 2.2.2). Note that this pre-processing step was done independently from the training of H2M.

### 2.1.3   Observational constraints

    Four observational hydrological variables were used to constrain H2M. The datasets were aggregated to a common spatial

resolution of $1°$ (Tab. 1). Due to differences in temporal coverage of the data products, a common period of February 2002 to December 2014 was selected.

    **i)** The monthly TWS observations from the Gravity Recovery and Climate Experiment (GRACE) Mascon Equivalent Water Height RL06 with Coastal Resolution Improvement (CRI) v1 (Watkins et al., 2015; Wiese et al., 2016, 2018) reflect vertically integrated variations in the water storage. These include the total variations of all storage components including groundwater,

soil moisture, surface water, biosphere-bound water, snow, and ice. To minimize the effect of outliers on the H2M performance, the TWS observations outside the range of -500 to 500 $\mathrm{mm}$ were excluded.

    **ii)** Monthly ET estimates were obtained from the global FLUXCOM-RS product (Tramontana et al., 2016; Jung et al., 2019), which is based on machine learning–driven estimates that are upscaled from site-level FLUXNET eddy covariance measurements (Baldocchi et al., 2001) to a global scale using a range of satellite-based drivers. ET was converted from latent

energy estimates assuming a constant latent heat of vaporization of 2.45 $\mathrm{MJ\,mm^{-1}\,m^{-2}}$.

    **iii)** Monthly Q estimates were obtained from the GRUN v1 dataset (Ghiggi et al., 2019). GRUN is based on an upscaling approach that correlates small catchment observations of Q to climate variability. The machine-learned relationships are then generalized to global scale. Note that only catchments with an area similar to the spatial resolution of the meteorological forcings were used for the prediction and thus, Q does not include larger routed streamflows and provides an estimate of

gridded runoff.

    **iv)** The daily SWE observations were obtained from the GlobSnow v2 product (Takala et al., 2011; Luojus et al., 2014). GlobSnow provides snow water equivalent in the Northern Hemisphere above $40°N$, while the mostly snow-free Southern Hemisphere is not covered. In GlobSnow, the time steps with no snow are encoded as missing values. Thus, we gap-filled the

**Table 2.** The terrestrial water storage (TWS) components as represented by the selected process models. While the hybrid hydrological model (H2M) represents snow water equivalent (SWE) explicitly, like the process models, the remaining TWS components are partitioned into soil cumulative water deficit (CWD) and groundwater (GW), which can be interpreted as fast and slow storage. To compare these components to the global hydrological models (GHMs), we calculated the storage as soil moisture plus canopy interception (CInt) if available and groundwater plus surface storage (SStor) if available, respectively. Note that CWD represents a *deficit* and thus, it corresponds to *negative* soil water storage.

|  | SWE | −CWD (fast storage) | | GW (slow storage) | |
|  |  | SM | CInt | GW | SStor |
| **Model** |  |  |  |  |  |
| LISFLOOD | ✓ | ✓ | ✗ | ✓ | ✗ |
| W3RA | ✓ | ✓ | ✗ | ✓ | ✗ |
| PCR-GLOBWB | ✓ | ✓ | ✓ | ✓ | ✓ |
| SURFEX-TRIP | ✓ | ✓ | ✓ | ✓ | ✓ |

SWE=soil water equivalent, CWD=cumulative soil water deficit, GW=groundwater,
SM=soil moisture, CInt=canopy interception, SStor=surface storage

GlobSnow product, but only with zero values if a) the snow cover fraction from MODIS (Hall and Riggs, 2016) was below
10 % and b) the GlobSnow product had missing values in a window of $\pm\,12$ d. The remaining missing values were not altered.

### 2.1.4   Global hydrological model ensemble

To evaluate the H2M simulations of TWS and its components, we selected the GHMs from the eartH2Observe ensemble (Schellekens et al., 2017), version WWR1. From the ten available model simulations, we selected those including groundwater storage: LISFLOOD (Van Der Knijff et al., 2010), W3RA (Van Dijk and Warren, 2010; Van Dijk et al., 2014), PCR-
GLOBWB (Van Beek et al., 2011; Wada et al., 2014), and SURFEX-TRIP (Decharme et al., 2010, 2013).

As the models represent different water storages (Tab. 2), they were combined to conceptually match storages modeled in the H2M (see Sect. 2.2.1): Snow water equivalent (SWE) is available in all models, and was used as is. Groundwater (GW) storage, conceptualized as all delayed storage components, is the sum of groundwater and surface storage (SStor), if available for a model. Soil moisture (SM) was combined with canopy interception (CInt), if available. Note that the H2M does
not represent SM directly but the cumulative soil water deficit (CWD), but we consider the dynamics of negative CWD to correspond to SM, and thus, the terms are used interchangeably when talking about soil moisture dynamics.

The GHMs were aggregated spatially from $0.5°$ to match the $1.0°$ resolution of our simulations. Such spatial aggregations for model comparison are common practice in model inter-comparison studies (e.g., Taylor et al., 2012). We expect the variations within four $0.5°$ cells to be small and thus assume that the $1.0°$ aggregation does not distort the modeled large-scale spatial
patterns.

### 2.1.5 Data filtering

The data used for H2M were additionally filtered to remove regions with low variations in the hydrological cycle, high anthropogenic impact, and with known data limitations using the following criteria:

1. Grid cells with more than 50 % water bodies, more than 90 % permanent snow or ice, or more than 90 % bare land.

2. Regions with more than 90 % artificial built-up surfaces.

3. Regions with large groundwater withdrawals labeled as "Groundwater depletion" under anthropogenic influence in Rodell et al. (2018).

4. Grid cells with more than 50 % missing values in any of the time series of the observational constraints.

5. Mountainous areas, which are masked in GlobSnow.

 After applying the filters, a total of 12 084 of 1°grid cells, covering roughly 80 % of the global land area, were selected.

## 2.2 The hybrid hydrological model (H2M)

The H2M consists of a dynamic neural network and a simple hydrological framework that represent the major water fluxes and changes in water storage (Fig. 1). The H2M is set up as a "global" model, i.e., the same model is used to predict the full spatio-temporal domain, in contrast to separate models for each grid cell in a "local" setup. The H2M only considers the
 vertical flow/transport of the water through the system and does not include the lateral flow of either surface (river routing) or sub-surface water (groundwater flow).

The neural network (Sect. 2.2.2) yields a set of time-varying coefficients conditioned on the meteorological forcing and spatial properties derived from the static input variables. These coefficients (e.g., snowmelt factor) are then used in a set of hydrological equations that are introduced in Sect. 2.2.1. For inference (after the optimization of the neural network), the model
 can be applied to unseen data like any forward simulation model without further model tuning.

For the sake of consistency and clarity, $\alpha$ denotes the time-varying coefficients that are directly estimated by the neural network, and $\beta$ denotes the global parameters that are learned as spatially constant. Throughout the manuscript, $t$ is used as time index and $i$ as the grid cell index. Uppercase variables are used for physical state variables. The code is available online (see "Code and data availability").

 ### 2.2.1 Hydrological components

In this section, we introduce the main hydrological components of the H2M.

#### Snow

Snow water equivalent is one of the water storages simulated by the H2M, and it is also constrained by the corresponding observation during model training.

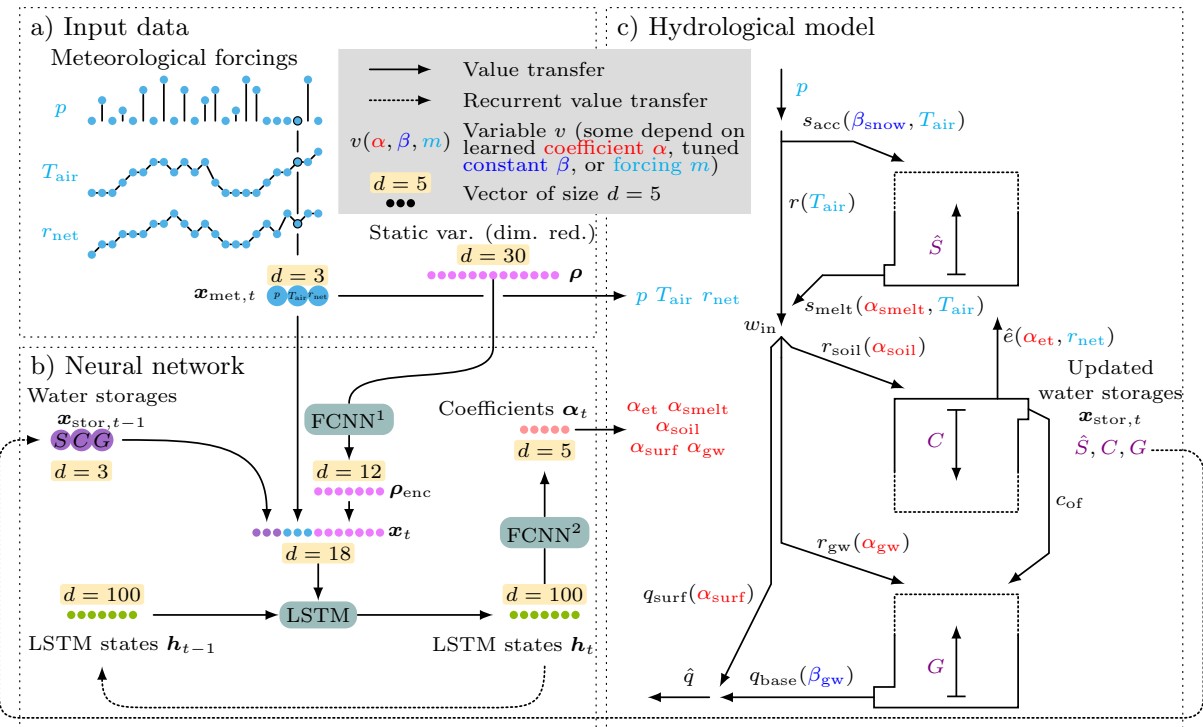

**Figure 1.** In the H2M, a (b) dynamic neural network (NN) simulates a set of time-varying coefficients that are used in a simple (c) hydrological model. The meteorological forcings $\boldsymbol{x}_{\mathrm{met},t}$ at time $t$ are used as input (a) to the NN and to the physical equations. The NN contains a long short-term memory (LSTM) layer and two fully connected networks (FCNN). The model maintains two sets of states: the (physical) water storages $\boldsymbol{x}_{\mathrm{stor}}$ and the LSTM's internal (non-physical) state $\boldsymbol{h}$ (cell state omitted here). It is conditioned on additional inputs representing static land surface and soil properties $\boldsymbol{\rho}$ and the previous water storages $\boldsymbol{x}_{\mathrm{stor},t-1}$. The NN module yields five time-varying coefficients ($\boldsymbol{\alpha}$) which are used in the balance equations. Two global parameters ($\boldsymbol{\beta}$) are estimated independently from the data input directly by the optimizer. The location of usage in the balance equations is indicated in parentheses, $(\hat{\cdot})$ denotes the variables that are constrained with observations, and upper case variables are storages. Forcings (cyan): $p$: precipitation, $T_{\mathrm{air}}$: air temperature, $r_{\mathrm{net}}$: net radiation. Water storages (purple): $\hat{S}$: snow water equivalent, $C$: cumulative soil water deficit, $G$: groundwater. Time-varying coefficients (red): $\alpha_{\mathrm{soil}}$: soil recharge fraction, $\alpha_{\mathrm{gw}}$: groundwater recharge fraction, $\alpha_{\mathrm{surf}}$: surface runoff fraction, $\alpha_{\mathrm{smelt}}$: snowmelt coefficient, $\alpha_{\mathrm{et}}$: evaporative fraction. Learned global constants (blue): $\beta_{\mathrm{snow}}$: snow undercatch correction constant, $\beta_{\mathrm{gw}}$: baseflow constant. Water fluxes: $r$: rainfall, $s_{\mathrm{acc}}$: snow accumulation, $s_{\mathrm{melt}}$: snowmelt, $w_{\mathrm{in}}$: liquid phase water input, $r_{\mathrm{soil}}$: soil recharge, $r_{\mathrm{gw}}$: groundwater recharge, $\hat{e}$: evapotranspiration, $c_{\mathrm{of}}$: overflow, $q_{\mathrm{surf}}$: surface runoff, $q_{\mathrm{base}}$: baseflow, $\hat{q}$: total runoff.

185  Snow accumulation

$$s_{\mathrm{acc},t,i} = p_{t,i} \cdot [T_{\mathrm{air},t,i} \leq 0] \cdot \beta_{\mathrm{snow}} \qquad \text{(in mm d}^{-1}\text{)} \qquad (1)$$

is precipitation $p$ with air temperatures $T_\text{air} \leq 0°\text{C}$. The accumulation is scaled by a learned (optimized) global constant $0 < \beta_\text{snow} < 1$. The correction accounts for the known overestimation of solid precipitation due to over-correction for under catch of snowfall in gauge measurements (Decharme and Douville, 2006). Potential snowmelt

$$s_{\text{melt},t,i} = \alpha_{\text{smelt},t,i} \cdot \max(T_{\text{air},t,i}, 0) \qquad (\text{in mm d}^{-1}) \qquad (2)$$

is then calculated using a degree-day approach. Opposite to snow accumulation, $s_\text{melt}$ occurs under the condition of $T_\text{air} > 0°\text{C}$. The time-varying snowmelt coefficient $\alpha_\text{smelt}$ is estimated by the neural network module and mapped to positive values by applying the softplus activation function; $\text{Softplus}(x) = \log(1 + e^x)$. The snow water equivalent

$$S_{t,i} = \max(S_{t-1,i} + s_{\text{acc},t,i} - s_{\text{melt},t,i}, 0) \qquad (\text{in mm}) \qquad (3)$$

is then updated using snow accumulation and melt. Positive values of $S$ are enforced by truncating negative values.

The temperature constraints on snowmelt and accumulation were introduced to avoid compensation effects between $s_\text{acc}$ and $s_\text{melt}$. It must be noted that such constraints are needed despite the fact that the relationship between snowfall or snowmelt and air temperature at 2 m may not always be realistic due to the corresponding associations with atmospheric (for snowfall) and land surface conditions (for snowmelt). We argue that the constraint will reduce or ideally remove equifinality among the parameters, and thus increase identifiability. This would allow for a physical interpretation of the parameters and processes.

**Soil recharge, groundwater recharge, and surface runoff**

The water input (in liquid form) $w_\text{in}$ ($\text{mm d}^{-1}$) is the sum of snowmelt and rainfall. It is partitioned into three fluxes: surface runoff, $q_\text{surf}$; soil recharge, $r_\text{soil}$; and groundwater recharge $r_\text{gw}$.

The coefficients for the partitioning are estimated by the neural network module and mapped to the range $(0, 1)$ and naturally constrained to the sum of 1 by applying the softmax transformation; $\text{Softmax}(x_j) = e^x_j / \sum_k^K e^x_k$ for the element $j$ of $K$ elements. The softmax transformation generalizes the logistic function to multiple dimensions. Note that the constrained training of parameters to 1 ensures that the incoming water is neither lost or generated during the partitioning respecting the physical law for the conservation of mass.

From the partitioning coefficients, soil recharge $r_\text{soil}$, groundwater recharge $r_\text{gw}$, and surface runoff $q_\text{surf}$ fluxes are then calculated as

$$r_{\text{soil},t,i} = \alpha_{\text{soil},t,i} \cdot w_{\text{in},t,i} \qquad (\text{in mm d}^{-1}), \qquad (4)$$

$$r_{\text{gw},t,i} = \alpha_{\text{gw},t,i} \cdot w_{\text{in},t,i} \qquad (\text{in mm d}^{-1}), \text{ and} \qquad (5)$$

$$q_{\text{surf},t,i} = \alpha_{\text{surf},t,i} \cdot w_{\text{in},t,i} \qquad (\text{in mm d}^{-1}), \qquad (6)$$

respectively, where $\alpha_\text{soil}$, $\alpha_\text{gw}$, $\alpha_\text{surf}$ are the partitioning coefficients of the total incoming water $w_\text{in}$. All partitioning parameters vary in both space and time.

### Evapotranspiration and soil moisture

The total evapotranspiration

$$e_{t,i} = \alpha_{\text{et},t,i} \cdot \frac{r_{\text{net},t,i}}{2.45} \qquad (\text{in mm d}^{-1}) \qquad (7)$$

is calculated as the product of the evaporative fraction $\alpha_{\text{et}}$ and net radiation $r_{\text{net}}$ (MJ d$^{-1}$ m$^{-2}$) converted to mm d$^{-1}$ assuming a latent heat of vaporization of 2.45 MJ mm$^{-1}$ m$^{-2}$. The evaporative fraction is learned by the neural network and mapped to the range $(0,1)$ by applying the sigmoid activation function; $\sigma(x) = 1/(1+e^{-x})$. Note that evapotranspiration is constrained by the corresponding observation during model training.

Once the evapotranspiration and soil recharge are calculated, the soil moisture is parameterized as the cumulative soil water deficit $C \geq 0$ as

$$C^*_{t,i} = C_{t-1,i} + r_{\text{soil},t,i} - e_{t,i} \qquad (\text{in mm}), \qquad (8)$$

$$c_{\text{of},t,i} = \text{Softplus}(C^*_{t,i}) \qquad (\text{in mm d}^{-1}), \text{ and} \qquad (9)$$

$$C_{t,i} = C^*_{t,i} - c_{\text{of},t,i} \qquad (\text{in mm}), \qquad (10)$$

which has the benefit of having a physical saturation limit of 0. For the comparison with the GHMs (Sect. 3.2), we calculate soil moisture (mm) dynamics as $M = -C$. The state $C$ is updated by addition of the soil recharge $r_{\text{soil}}$, subtraction of evapotranspiration $e$ (Eq. 8), and leveling by the overflow mechanism (Eq. 9–10): If $C$ approaches 0, an overflow mechanism allows for direct discharge of excess soil moisture into the deeper groundwater storage. Due to the heterogeneity within a model cell, the overflow $c_{\text{of}}$ starts already at values close to 0, which is achieved by using the softplus function.

### Baseflow and Groundwater

The baseflow

$$q_{\text{base},t,i} = G_{t-1,i} \cdot \beta_{\text{gw}} \qquad (\text{in mm d}^{-1}) \qquad (11)$$

is calculated as fraction of the past groundwater storage $G_{t-1}$ via the learned global baseflow constant $\beta_{\text{gw}}$ with the range $(0,1)$. Once the baseflow, groundwater recharge, and overflow of soil storage are calculated, the groundwater storage

$$G_{t,i} = G_{t-1,i} + c_{\text{of},t,i} + r_{\text{gw},t,i} - q_{\text{base},t,i} \qquad (\text{in mm}) \qquad (12)$$

can be updated using a simple water balance. In H2M, $G$ represents an unconfined aquifer with an unlimited storage capacity.

### Total Runoff

The total runoff

$$q_{t,i} = q_{\text{surf},t,i} + q_{\text{base},t,i} \qquad (\text{in mm d}^{-1}) \qquad (13)$$

is simply calculated as the sum of the surface runoff $q_{\text{surf}}$ (Eq. 6) and the baseflow $q_{\text{base}}$ (Eq. 11). We emphasize here that the neural network receives the state of water storage as inputs and is, thus, able to learn interactions of the water storages, the input variables, and the corresponding hydrological partitioning and outflow coefficients. Thus, the runoff generation and evapotranspiration processes do not only depend on the current and past meteorological condition and static variables, but also on hydrological state, e.g., the soil water deficit. Therefore, we additionally use runoff as a data constraint during model training.

**H2M storage components**

For model training against GRACE, the variations of the modeled terrestrial water storage components are added to calculate the total terrestrial water storage

$$T_{t,i}^* = S_{t,i} + G_{t,i} + (-C_{t,i}) \qquad\qquad (\text{in mm}). \qquad\qquad (14)$$

Note that $-C$ is used in Eq. 14 as $C$ itself is defined as the water *deficit*. As the observations of the terrestrial water storage from GRACE represent the temporal *variations*, the mean of simulated storage were removed from each grid cell as

$$T_{t,i} = T_{t,i}^* - \frac{1}{\mathcal{T}} \cdot \sum_{k=1}^{\mathcal{T}} T_{k,i}^* \qquad\qquad (\text{in mm}), \qquad\qquad (15)$$

where $k$ is the time step of $\mathcal{T}$ total steps. The TWS is constrained by observations during model training.

Note that H2M does not represent surface water storage—a fourth major component of TWS, dominant especially in and around large surface water bodies like rivers and lakes—explicitly. This will be considered in the discussion of the results.

Compared to physically-based models, the H2M does not explicitly partition the sub-surface storages as soil moisture and groundwater storages. Rather, it is represented as GW and CWD. The partition is an emergent behavior of H2M constraints by the major hydrological fluxes. Negative CWD is loosely and conceptually interpreted as root zone soil moisture, as it serves as the moisture source for evapotranspiration. This is in fact consistent with the physical models, even though CWD does not have a continuous interaction with GW storage except during overflow in H2M.

GW storage represents all delayed residual liquid water storage with infinite capacity. It is constrained by the baseflow fraction and subsequently temporal variation of total runoff (Eq. 11), which leads to a delayed dynamics compared to CWD.

### 2.2.2  The neural network (NN) module

The NN module (Fig. 1b) consists of three consecutively arranged sub-modules employed for extractions of different features. Overall, the NN module learns spatio-temporally varying coefficients of the hydrological model using meteorological and dimensionality-reduced static variables of land (sub)surface characteristics. The pseudo-code of the NN module is presented in Appendix E, while the sub-modules are introduced here.

The first feed-forward (i.e., non-temporal) sub-module learns a compressed representation of the static variables (Eq. 16). This representation, together with meteorological input, is then fed into the second sub-module, a recursive long short-term

memory (LSTM) model (Hochreiter and Schmidhuber, 1997), shown in Eq. 17. The third sub-module (Eq. 18) transforms the outputs of the LSTM to a set of coefficients, which are then fed into the hydrological components. As the model weights are shared across all grid cells, the NN module learns from the global dynamics and not exclusively from each grid cell. For a comprehensive overview of the neural network architectures, see Goodfellow et al. (2016).

The first sub-module

$$\boldsymbol{\rho}_{\text{enc},i} = f_{\text{FCNN}^1}(\boldsymbol{\rho}_i) \tag{16}$$

is a fully-connected neural network (FCNN[1] in Fig. 1) with a single hidden layer and 150 nodes. It takes the static encodings $\boldsymbol{\rho}$ (see Sect. 2.1.2) as inputs and transforms them into a more condensed form ($\boldsymbol{\rho}_{\text{enc}}$). This reduces the high dimensionality of static inputs from 30 to 12 values. Ideally, this lower-dimensional representation describes the most significant gradients of the land characteristics at the sub-grid scale (visualized in Fig. C2, Appendix C). Note that the static variables have already been compressed in a pre-processing step, and the transformation in this sub-module is optimized specifically for the parameterization of the hydrological components.

The second sub-module is an LSTM, a recurrent neural network (RNN) variant that updates its states dynamically using the previous states and the current input. LSTMs are broadly used in the Earth sciences due to their ability to learn temporal dynamics (Körner and Rußwurm, 2021), i.e., to represent memory effects that are present in hydrological observations (Kraft et al., 2019, 2021; Humphrey et al., 2016). It has a hidden (in the sense of "latent") state vector $\boldsymbol{h}$ whose length (100 in H2M) is a tunable hyper-parameter. The hidden state

$$\boldsymbol{h}_{i,t} = f_{\text{RNN}}(\boldsymbol{h}_{i,t-1}, \boldsymbol{x}_{t,i}) \tag{17}$$

is updated at each time step by using interactions of the previous states $\boldsymbol{h}_{t-1}$ and the current input $\boldsymbol{x}_{t,i}$. In H2M, $\boldsymbol{x}_{t,i}$ is a multivariate input consisting of concatenated current meteorological conditions $\boldsymbol{x}_{\text{met},t,i}$, antecedent physical states from the hydrological model $\boldsymbol{x}_{\text{stor},t-1,i}$, and the static features $\boldsymbol{\rho}_{\text{enc},i}$ from Eq. 16. The input allows the LSTM to learn interactions among the variables conditioned on static land properties like land cover type or elevation. In the optimization process, the RNN learns to maintain a memory of information from past time steps and is capable of updating, removing, and extracting information from its state.

In summary, the LSTM sub-module is similar to a physically-based model—it takes the current inputs and static characteristics, and updates the system state based on their interactions with the past state. It should be noted that neither its hidden state nor the update function is physically interpretable.

Lastly, the third sub-module

$$\boldsymbol{\alpha}_{t,i} = f_{\text{FCNN}^2}(\boldsymbol{h}_t) \tag{18}$$

linearly maps the LSTM output $\boldsymbol{h}_t$ to the coefficients $\boldsymbol{\alpha}$ of the hydrological components (FCNN[2] in Fig. 1). The vector $\boldsymbol{\alpha}$ contains five time-varying scalars corresponding to soil recharge fraction $\alpha_{\text{soil}}$, groundwater recharge fraction $\alpha_{\text{gw}}$, surface runoff fraction $\alpha_{\text{surf}}$, snowmelt coefficient $\alpha_{\text{smelt}}$, and evaporative fraction $\alpha_{\text{et}}$.

## 2.3  Model training

This section introduces the necessary aspects of the model training and validation. First, we introduce the cross-validation setup, followed by the model training and the loss function.

### 2.3.1  Cross-validation setup

We use $k$-fold cross-validation to validate the H2M against observations that were withheld during the training. In the cross-validation, the model is optimized first on a set of training grid cells and applied to a different set of test grid cells, i.e., spatial splitting. Specifically, the grid cells were first split into four sets of grids $g_l, l \in \{1, 2, 3, 4\}$, each consisting of every second grid cell in latitude and longitude direction with an offset $O_l$. The offsets of $O = \{(0,0), (0,1), (1,0), (1,1)\}$ are chosen such that the selected grids did not overlap while covering the full spatial domain. This procedure asserts a minimum distance needed to avoid potential issues of spatial autocorrelation (Roberts et al., 2017) *within each grid*. Each grid was then randomly subdivided into five folds for cross-validation: three folds for training, and one each for validation and testing. The validation subset was used in early stopping, i.e., to stop the training after the validation loss increases over several consecutive iterations. After the training stop, the best model parameters are loaded and predictions are made on the test subset which are used as the final prediction. In the iteration through the folds, every fold is used once in the test set, and as such, a complete set of predictions for a grid cell that was not informed by its own observation is obtained for the respective grid.

In addition to the spatial splitting, the data were also split into calibration and validation time periods akin to the traditional approach. To do so, February 2002 to December 2008 was used for calibration, and January 2009 to December 2014 was used for validation and testing.

The hyper-parameters of the NN (i.e., the number of layers and hidden nodes in the neural networks, the learning rate, weight decay, dropout, and gradient clipping) are determined on a single grid, and the cross-validation is only applied on the remaining three grids. For hyper-parameter tuning, we employed the Bayesian optimization hyper-band (BOHB) algorithm (Falkner et al., 2018) as implemented in the *ray.tune* framework (Liaw et al., 2018).

This setup was chosen to avoid over-fitting, which is needed due to the data adaptivity of neural networks. Note that the spatial splitting reduces the dependency between the cross-validation sets, but does not completely remove it. In addition to the spatial and temporal splitting and the early stopping, we used weight decay (Loshchilov and Hutter, 2017) for regularization.

### 2.3.2  Training setup

As the neural networks and the hydrological equations are differentiable, standard gradient descent approaches with back-propagation can be used for optimizing the H2M (Goodfellow et al., 2016). We use a multi-task loss as optimization objective which is a recent concept in deep learning for multi-criteria model calibration (see below), and *AdamW* (Loshchilov and Hutter, 2017) as the optimizer.

Following a common practice in machine learning, the input variables and the observational data constraints are each $z$-transformed individually to follow a standard normal distribution using the pre-computed mean and standard deviations from

the training set. For physical consistency, the corresponding non-transformed variables are used for the hydrological balance equations (see Sect. 2.2).

To obtain an equilibrium of physical and hidden states of H2M, a model spin up is carried out with spin up data of five years duration, with each full year selected randomly from the training set. In each optimization iteration, the model is first forced by the spinup data to retrieve steady states, which are then used as initial conditions during the full forward run with parameter updates (see pseudo-code in Appendix E).

### 2.3.3 Multi-task loss

The goal of the model optimization is to minimize the total loss, which consists of two major aspects:

1) The loss term

$$\mathcal{L}_v(\boldsymbol{x}, \boldsymbol{y}; \boldsymbol{\phi}, \boldsymbol{\beta}) = \sum_{t=1}^{\mathcal{T}} \sum_{i=1}^{\mathcal{I}} ||y_{v,t,i} - \hat{y}_{v,t,i}||^2 \quad , v \in \{T, S, e, q\} \tag{19}$$

is calculated as the sum of squared residuals for each $z$-transformed observational data constraint. Here, $y_{v,t,i}$ and $\hat{y}_{v,t,i}$ are the observed and predicted values of the variable $v$, respectively. The predictions depend on the input data $\boldsymbol{x}$, the neural network parameters $\boldsymbol{\phi}$, as well as the learned global constants $\boldsymbol{\beta}$. An additional loss term is employed to promote parameters that would lead to near zero cumulative soil water deficit $C$ (soil becomes saturated) at least occasionally:

$$\mathcal{L}_C(\boldsymbol{x}; \boldsymbol{\phi}, \boldsymbol{\beta}) = \sum_{t=1}^{\mathcal{T}} \sum_{i=1}^{\mathcal{I}} (p_{10}(\hat{C}_{t,i}) + b_{\mathrm{c}}) \cdot w_{\mathrm{c}} \quad . \tag{20}$$

This term pushes the lower 10 percentile $p_{10}$ of $C$ towards zero. It was needed to reduce the state drift mostly related to spinup with random years of data that resulted in non-interepretable offsets in $C$ (Kraft et al., 2020). A bias $b_{\mathrm{c}} = 0.1$ was added to prevent the loss from becoming zero, which would interfere with the multitask loss weighting described below. The loss weight $w_{\mathrm{c}}$ was lowered consecutively during training such that the loss $\mathcal{L}_{\mathrm{C}}$ had only an impact during the early training phase.

2) A task uncertainty term $\boldsymbol{\sigma}$, weighting the individual losses dynamically:

$$\mathcal{L}_{\mathrm{total}}(\boldsymbol{x}, \boldsymbol{y}; \boldsymbol{\phi}, \boldsymbol{\beta}, \boldsymbol{\sigma}) = \sum_{v \in \{T, S, e, q, C\}} \frac{1}{2 \cdot \sigma_v^2} \mathcal{L}_v + \log(\sigma_v) \quad , \tag{21}$$

where $\boldsymbol{\sigma}$ is a vector of task-specific uncertainties used to give more or less weight to a particular loss term. The task-specific uncertainties are trained during optimization such that the emphasis on a specific task changes dynamically over the course of the model optimization. Note that $\log(\sigma_v)$ prevents the uncertainties from diverging to infinity. This approach, called *self-paced multi-task weighting* (Kendall et al., 2018), is advantageous as the weights do not need to be subjectively predefined. The weights are visualized in Fig. C1, Appendix C.

Hence, the global optimization problem can be expressed as

$$\boldsymbol{\theta}^* = \underset{\boldsymbol{\theta} = (\boldsymbol{\phi}, \boldsymbol{\beta}, \boldsymbol{\sigma})}{\arg \min} \mathcal{L}_{\mathrm{total}}(f_{\boldsymbol{\phi}, \boldsymbol{\beta}}, \boldsymbol{x}, \boldsymbol{y}, \boldsymbol{\sigma}) \quad , \tag{22}$$

in which the parameters of the neural network $\boldsymbol{\phi}$, the global constants $\boldsymbol{\beta}$, and the task weights $\boldsymbol{\sigma}$ are all concurrently and simultaneously optimized.

## 2.4 Model evaluation and analysis

This section introduces the performance metrics, the spatial and temporal scales, and the methods used to decompose the TWS components.

### 2.4.1 Performance metrics

The quality of the model predictions was mainly assessed using the Nash–Sutcliffe model efficiency coefficient (NSE)

$$e_{\text{NSE}} = 1 - \frac{\sum_{i=1}^{N} (m_i - o_i)^2}{\sum_{i=1}^{N} (o_i - \bar{o})^2} \quad , \tag{23}$$

where $m_i$ is the modeled and $o_i$ the observed value, $N$ is the total number of data points, and $\bar{o}$ is the mean of observations (Nash and Sutcliffe, 1970). An NSE of $e_{\text{NSE}} = 1$ indicates a perfect fit, while an NSE of $e_{\text{NSE}} = 0$ ($e_{\text{NSE}} < 0$) indicates that the predictive performance of the model is the same as (worse than) that of the mean. Additionally, the root mean square error (RMSE), the Pearson correlation coefficient ($r$), and the ratio of modeled and observed standard deviation (SDR) were used for model performance evaluation.

### 2.4.2 Temporal and spatial scales

The performance of H2M was evaluated across different temporal scales. To do so, the observed and modeled time series were decomposed into the mean seasonal cycle (MSC) and the interannual variability (IAV) as

$$v_{\text{MSC},m} = \frac{1}{Y} \sum_{y=1}^{Y} v_{m,y} \quad , \text{ and} \tag{24}$$

$$v_{\text{IAV},m,y} = v_{m,y} - v_{\text{MSC},m} \quad , \tag{25}$$

where $v$ is the observed or modeled time series, $m$ is the month, and $y$ is the year out of $Y$ total years. Before calculating the model performance metrics for MSC and IAV, the linear trends were removed from the time series.

Spatially, the model performance is also evaluated across several scales to investigate robustness of the model for local to global scale variations. For the regional-scale analysis, we use continent-wise hydroclimatic biomes from Papagiannopoulou et al. (2018), a machine learning–based dataset that accounts for climate-vegetation interactions. The number of classes was reduced by combining some of the similar sub-regions, e.g., transitional water-driven and transitional energy-driven or subtypes of boreal regions (Fig. 2). While aggregating the modeled variables to a regional scale, an area-weighted method was used to accommodate for differences in the grid-area across the latitude.

For the global-scale performance we calculate the metrics in two different ways that produce a single metric by a mapping function $f_{\text{perf}} : \mathbb{R}^{\mathcal{T}} \times \mathbb{R}^{\mathcal{T}} \mapsto \mathbb{R}$ that compares two sequences of length $\mathcal{T}$. The first, which we call the *global performance*

$$\mathcal{M}_{\text{global}} = f_{\text{perf}}\big(\{\mu_{\text{m},t}\}_{t=1,\dots,\mathcal{T}}, \{\mu_{\text{o},t}\}_{t=1,\dots,\mathcal{T}}\big) \tag{26}$$

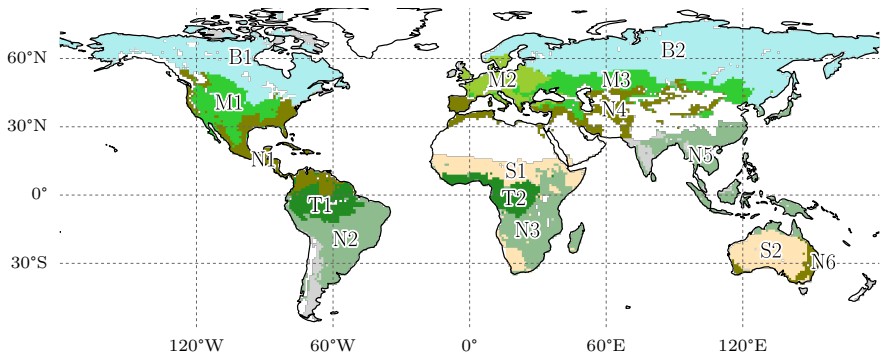

**Figure 2.** Continental hydro-climatic regions, adapted from Papagiannopoulou et al. (2018). **Boreal**: North America (B1) and Eurasia (B2). **Temperate**: North America (M1), Europe (M2), and Asia (M3). **Transitional**: North and Central America (N1), South America (S2), Africa (N3), Eurasia and North Africa (N4), Southeast Asia (N5), and Australia (N6). **Subtropical**: Africa (S1) and Australia (S2). **Tropical**: South America (T1) and Africa (T2).

represents the performance of the globally-aggregated variables. The variables $\mu_{\mathrm{m},t}$ and $\mu_{\mathrm{o},t}$ represent the modeled and the observed weighted spatial mean for one time step $t$, respectively. Similar to regional-scale evaluations, these metrics reflect how the area-weighted globally aggregated time-series compare. The global-scale signal are themselves useful indicators, as they are often used to characterize the Earth system and land surface processes, e.g., climatic changes (Pachauri et al., 2014), or to evaluate water-carbon relations (Jung et al., 2017; Humphrey et al., 2016).

In contrast, global summary of the *local performance*

$$\mathcal{M}_{\mathrm{local}} = \mathrm{median}\big(\big\{ f_{\mathrm{perf}}(m_{t,i}, o_{t,i})\big\}_{i=1,\dots,\mathcal{I}}\big) \tag{27}$$

is indicative of how the model performs locally all over the globe. Here, the performance is first calculated for the modeled ($m$) versus observed ($o$) time series per grid cell $i$. The resulting cell-wise metric is then reduced using the area-weighted median. The local metrics are useful because the positive and negative model errors and tendencies can compensate when aggregated over a large spatial extent (e.g., Jung et al., 2017).

### 2.4.3 Terrestrial water storage variations and decomposition

For the analysis on the decomposition of TWS (Sect. 3.4 and Sect. 4.2.2), we use the simulated variables SWE, GW, and CWD to assess their contributions to the TWS dynamics, seasonality, and interannual variability. Note that CWD represents a *deficit* of water in the soil. As a consequence, CWD shows opposite dynamics to water storages. We calculate the absolute

$$\mathcal{A}_v = \sum_{t=1}^{\mathcal{T}} |v_t - \bar{v}|, \quad v \in \{-C, G, S\} \tag{28}$$

and relative contribution (hereinafter simply *contribution*)

$$C_v = \frac{\mathcal{A}_v}{\sum_{w \in \{-C, G, S\}} \mathcal{A}_w} \quad v \in \{-C, G, S\} \tag{29}$$

for each component $v$ following Getirana et al. (2017). Here, $\bar{v}$ is the mean over the time series $v$. The contributions are calculated per grid cell for the time series and their MSC and IAV.

## 3 Results

We first assess the performance of H2M simulations against the four observational data constraints (TWS, SWE, Q, and ET) at different spatial and temporal scales. This is followed by a comparison and benchmarking of model performance of H2M TWS and SWE against the simulations from four GHMs in the eartH2Observe ensemble. As the hybrid modeling framework has been significantly developed since Kraft et al. (2020), the H2M performance needs to be re-evaluated here. After the evaluations, we present a closer analysis and interpretation of the parameters estimated by the neural network that define the
hydrological responses and generation of key hydrological fluxes in H2M. Finally, we present and compare the partitioning of TWS components.

An optimization run of a single cross-validation iteration takes 6 hours, a forward run for all grid-cells and the entire period from 2002 to 2014 takes about 15 minutes. Each model was run on a NVIDIA Tesla Volta V100 16 GB GPU with up to 10 CPUs (Intel(R) Xeon(R) @ 2.20GHz) for data buffering and background tasks.

### 3.1 General model performance

For the assessment of the H2M performance, we only used grid cells from the test set and time steps from the test period of 2009 to 2014, which were not used during the model training, and hence not seen by the neural network component of H2M.

The model reproduced the patterns of the observed variables well (Tab. 3). In general, the global signal (global performance, see Eq. 26) was reproduced better than the local cell-level signal (local performance, see Eq. 27). For both observational
constraint variables TWS and SWE an NSE $e_{\mathrm{NSE}} > 0.8$ and Pearson's correlation $r > 0.9$ on the global and $e_{\mathrm{NSE}} > 0.5$ and $r > 0.8$ for the local level was achieved. The seasonal signals of $\mathrm{TWS_{MSC}}$ and $\mathrm{SWE_{MSC}}$ were modeled with high accuracy ($e_{\mathrm{NSE}} > 0.9$ on global, $e_{\mathrm{NSE}} = 0.7$ on local level) while the interannual variability performance varied: The $\mathrm{TWS_{IAV}}$ was reproduced well with $e_{\mathrm{NSE}} = 0.54$ ($r = 0.8$) on global, and with $e_{\mathrm{NSE}} = 0.26$ ($r = 0.67$) on local level. The $\mathrm{SWE_{IAV}}$ performance was decent for the global signal ($e_{\mathrm{NSE}} = 0.22$, $r = 0.87$), but lower ($e_{\mathrm{NSE}} = 0.15$, $r = 0.64$) on local level.

Both ET and Q are machine learning model–based and not directly observed at global scale. The patterns were reproduced well in terms of the seasonality on the global level, while the local performance was lower. For the $\mathrm{ET_{IAV}}$, a low NSE ($e_{\mathrm{NSE}} = -0.17$) on global, and on cell-level ($e_{\mathrm{NSE}} = -0.65$) is achieved, while the correlation is still relatively good with $r = 0.67$ on global, and $r = 0.6$ on local level. The SDR, the ratio of modeled and observed standard deviation, indicates that on both global and local level the variability of the simulated $\mathrm{ET_{IAV}}$ signal is substantially larger than the reference data with SDR of 1.41 on global, and SDR of 1.65 on cell-level (see Fig. A2 in the Appendix for spatial patterns). For Q, the performance is decent

**Table 3.** The global (spatially averaged) and local (median cell-level) model performance for the observational constraint variables terrestrial water storage (TWS), snow water equivalent (SWE), evapotranspiration (ET), and runoff (Q), and their decomposition into the mean seasonal cycle (MSC) and interannual variability (IAV). The Nash–Sutcliffe model efficiency (NSE), Pearson correlation (r), root mean square error (RMSE), and the ratio of modeled and observed standard deviation (SDR) are calculated for the test set, values represent the mean across the 15 cross-validation runs. Positive values of SDR indicate that the modeled variance is larger than the observed. Note that for the SWE, cells with constant 0 were dropped. The values were calculated for the test set in the range 2009 to 2014 on monthly time scale.

| | Metric | TWS | MSC | IAV | SWE | MSC | IAV | ET | MSC | IAV | Q | MSC | IAV |
|---|---|---|---|---|---|---|---|---|---|---|---|---|---|
| Global performance | NSE (-) | 0.84 | 0.93 | 0.54 | 0.96 | 0.96 | 0.22 | 0.96 | 0.96 | -0.11 | 0.75 | 0.78 | 0.47 |
| | Pearson's $r$ (-) | 0.94 | 0.97 | 0.80 | 0.98 | 0.98 | 0.87 | 1.00 | 1.00 | 0.67 | 0.93 | 0.97 | 0.81 |
| | SDR (-) | 1.15 | 1.10 | 1.09 | 1.02 | 1.01 | 1.57 | 0.99 | 0.99 | 1.41 | 0.93 | 0.87 | 1.13 |
| | RMSE (mm) | 7.33 | 4.97 | 3.27 | 5.22 | 5.98 | 2.16 | 0.07 | 0.07 | 0.02 | 0.06 | 0.05 | 0.03 |
| Local performance | NSE (-) | 0.54 | 0.70 | 0.26 | 0.58 | 0.74 | 0.15 | 0.79 | 0.87 | -0.77 | 0.20 | 0.17 | 0.07 |
| | Pearson's $r$ (-) | 0.82 | 0.93 | 0.67 | 0.89 | 0.96 | 0.64 | 0.95 | 0.98 | 0.60 | 0.80 | 0.91 | 0.62 |
| | SDR (-) | 0.98 | 1.09 | 0.95 | 0.91 | 0.92 | 0.97 | 1.03 | 1.01 | 1.65 | 0.98 | 0.97 | 1.04 |
| | RMSE (mm) | 42.80 | 22.59 | 28.72 | 15.49 | 13.13 | 10.60 | 0.27 | 0.22 | 0.14 | 0.44 | 0.31 | 0.27 |

on the global level and lower on the local cell-level. Also here, low values in terms of NSE are accompanied by relatively good correlation. Because the independent data for ET and Q are not direct observations, we focus on TWS and SWE in the following. Maps of mean simulated versus observed fluxes and the spatial patterns of the model performance are provided in Appendix A.

## 3.2 Benchmarking H2M against GHMs

For the quantitative benchmarking of H2M performance with the state-of-the-art GHMs from eartH2Observe (see Sect. 2.1.4), we use the common time period of 2009 to 2012 (not 2009-2014 as in the previous section) but all common grid cells between the GHMs and H2M. This is justified as H2M has a negligible generalization error in space, i.e., the H2M performance is not systematically better in training grid cells. Similarly, we use the entire common time period (including the training data) for the *qualitative* assessment of the water cycle dynamics, as also in time, the generalization error was small. We note here that H2M was optimized with the datasets used for evaluation, while the GHMs have either been calibrated using catchment-level observational runoff data (LISFLOOD) or rely on prior parameter estimation (W3RA, SURFEX-TRIP, PCR-GLOBWB) alone (Schellekens et al., 2017). The comparison presented here serves the purpose of performance benchmarking of the hybrid modeling approach rather than finding the "best" model.

The H2M modeling efficiency (i.e., the NSE) falls within the range of the GHMs in terms of the global performance (⋄ in Fig. 3), although the performance varies less across the variables and temporal scales. However, H2M achieves a consistently

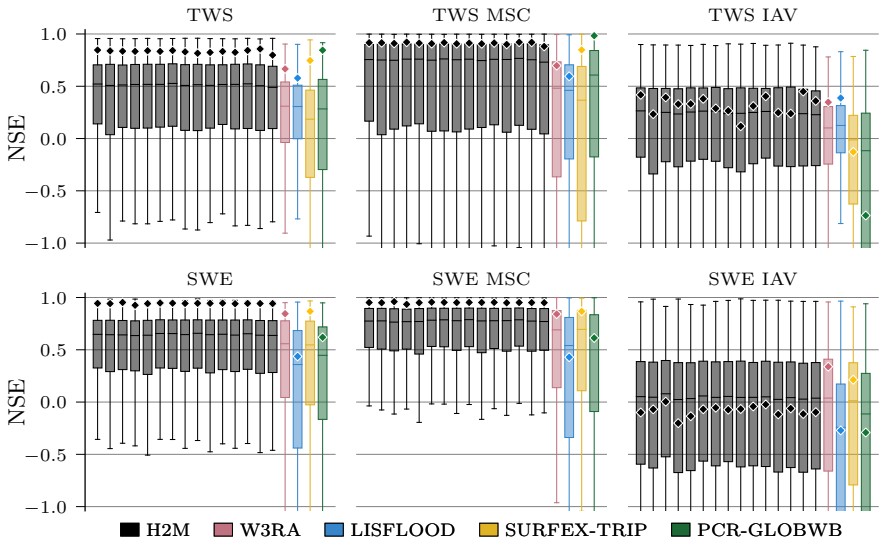

**Figure 3.** Global and local grid cell–level Nash–Sutcliffe model efficiency coefficient (NSE) of the hybrid hydrological model (H2M) and the process-based global hydrological models (GHMs) for the terrestrial water storage (TWS) on top and the snow water equivalent (SWE) at the bottom. The gray bars represent individual cross-validation runs. The ⋄-markers show the global (spatially averaged signal) model performance, the boxes represent the spatial variability of the local cell-level performance. The y-axis was cut at -1 due to some large negative NSE values. The panels show the model performance in respect to the full-time series, the mean seasonal cycle (MSC), and the interannual variability (IAV). Note that for SWE, only grid cells with at least one day of snow are shown, as the NSE is not defined if the observations are constant zero, which would lead to a comparison of different grid cells. The metrics are calculated from the complete common time range from 2009 to 2012 on monthly time scale. Note that deviations from the numbers reported in Tab. 3 are due to different time ranges.

higher local performance (boxes in Fig. 3). The TWS is reproduced slightly better by the PCR-GLOBWB, which, however, has a relatively low performance on the local scale. All models struggle to reproduce the $SWE_{IAV}$ signal: The median NSE of H2M is on a par with W3RA and SURFEX-TRIP, while the performance on spatially aggregated level is lower. A comparison of the model performance using the same forcings as in the eartH2Observe ensemble is provided in Appendix D, Fig. D1.

While all models reproduce the global monthly and seasonal TWS (Fig. 4) relatively well, the results vary more substantially for the $TWS_{IAV}$. Here, the H2M, WR3A, and LISFLOOD models show the best agreement with the TWS observations (also see Fig. 3 of model performance). The lower agreement of SURFEX-TRIP and PCR-GLOBWB on the global interannual scale can be attributed to the time periods 2005–2006 and 2008–2010, respectively. From Fig. B1 of the regional averages (Appendix B), it becomes evident that this low agreement on global level is mainly due to a low agreement in the tropical regions (T1: S-AM tropical and T2: AFR tropical).

The global SWE was well reproduced by H2M, especially the seasonal cycle showed better agreement than the GHMs, where the latter agreed well with the timing, but not the magnitude (Fig. 5). The global interannual variability was not reproduced

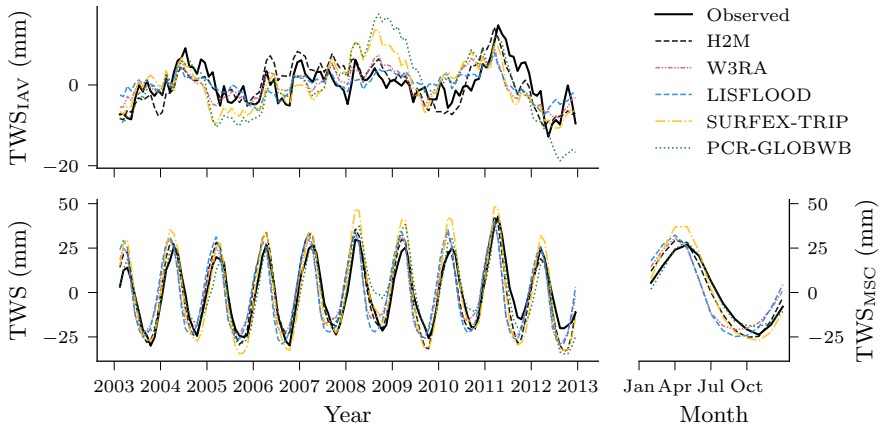

**Figure 4.** Comparison of the hybrid hydrological model (H2M) and a set of process-based global hydrological models (GHMs) of the terrestrial water storage (TWS), its mean seasonal cycle (TWS$_{MSC}$) and its interannual variability (TWS$_{IAV}$) for the global signal. The time series were aggregated using the cell size weighted mean across all grid cells. The regional time series are show in Appendix B, Fig. B1.

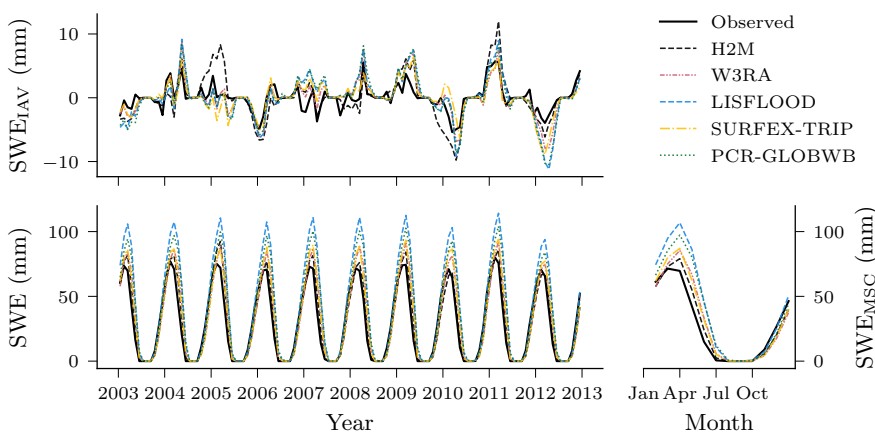

**Figure 5.** Comparison of the hybrid hydrological model (H2M) and a set of process-based global hydrological models (GHMs) of the snow water equivalent (SWE), its mean seasonal cycle (SWE$_{MSC}$) and its interannual variability (SWE$_{IAV}$) for the global signal. The time series were aggregated using the cell size weighted mean across all grid cells. The regional time series are show in Appendix B, Fig. B2.

well by the H2M, LISFLOOD, and PCR-GLOBWB. Interestingly, H2M performed best when forced by the same WFDEI data
as in the GHM simulations (Fig. D1 in Appendix D). Regional model comparison of the time series are provided in Fig. B1 and B2, Appendix B.

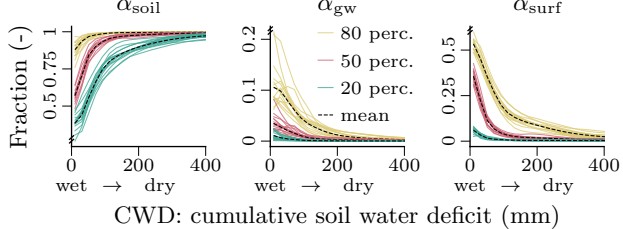

**Figure 6.** Relationship between the water input partitioning fractions for soil ($\alpha_{\text{soil}}$, left), groundwater ($\alpha_{\text{gw}}$, middle), and surface runoff ($\alpha_{\text{surf}}$, right), and the cumulative soil water deficit (CWD) as learned by the neural network. The figure shows the respective percentiles of the spatio-temporal conditional distribution $P(\alpha \mid C \in B_i)$, where $C$ is the cumulative soil water deficit on the x-axis discretized into $N = 10$ bins $B = \left\{ [0, 40), \ldots, [360, 400) \right\}_{i=1,\ldots,N}$. The colored lines show the percentiles per cross-validation run, the black dashed lines show the mean across the colored lines. The CWD dynamics correspond to negative soil moisture, i.e., larger CWD for dryer soils, and thus a larger CWD corresponds to smaller soil moisture. The plots are based on global daily cell time steps from 2009 to 2014. Note the differences in y-scale.

### 3.3 Hydrological responses in H2M

For the qualitative assessment of the hydrological responses, we use all grid cells, like in the previous section, and show the time range from 2003 to 2014 in time series plots. This involves the training data, but the impact is minimal due to a
negligible generalization error. The H2M yields a set of data-driven, spatio-temporally varying coefficients that define the hydrological responses and generation of key hydrological fluxes. In particular, we focus on four parameters: $\alpha_{\text{soil}}$, the fraction of throughfall that percolates into the soil; $\alpha_{\text{gw}}$, the fraction that recharges the groundwater, $\alpha_{\text{surf}}$; the fraction that runs off as surface runoff component; and $\alpha_{\text{et}}$, the evaporative fraction (ratio of evapotranspiration to net radiation). Here, we analyze the spatio-temporal variability of the parameters and how they are associated with soil moisture condition defined by soil water
deficit. In essence, these are analogous to stage-discharge relationships (Kumar, 2011) that are commonly used to characterize hydrological responses of river discharge at the catchment scale.

    The partitioning of the liquid water input $w_{\text{inp}}$ (rainfall plus snowmelt) by the fractions for soil recharge ($\alpha_{\text{soil}}$), groundwater recharge ($\alpha_{\text{gw}}$), and surface runoff ($\alpha_{\text{surf}}$) was robust across cross-validation runs and showed a clear relationship to CWD (Fig. 6). With an increasing soil water deficit (larger CWD, dryer soil), the soil recharge increases, while the groundwater
recharge and surface runoff decrease. For a CWD below 200 mm, we observe a large spatio-temporal variation in the partitioning, evident through the relatively large difference between the 20[th] and 80[th] percentiles. The transition from larger soil recharge to larger groundwater recharge and surface runoff is exponentially decreasing, i.e., the change is faster with lower CWD (wetter soil). Above a CWD of 200 mm (dry soil), the partitioning is constant in space and time with $\alpha_{\text{soil}}$ converging to 1, while $\alpha_{\text{gw}}$ and $\alpha_{\text{surf}}$ converge to 0. The relatively large variation under wet conditions (low CWD) in Fig. 6 can be attributed about equally
to temporal and spatial variability. The groundwater recharge fraction $\alpha_{\text{gw}}$ shows a slightly larger temporal variability than the other fractions, and the contribution of the temporal component was generally a bit lower in the transitional regions.

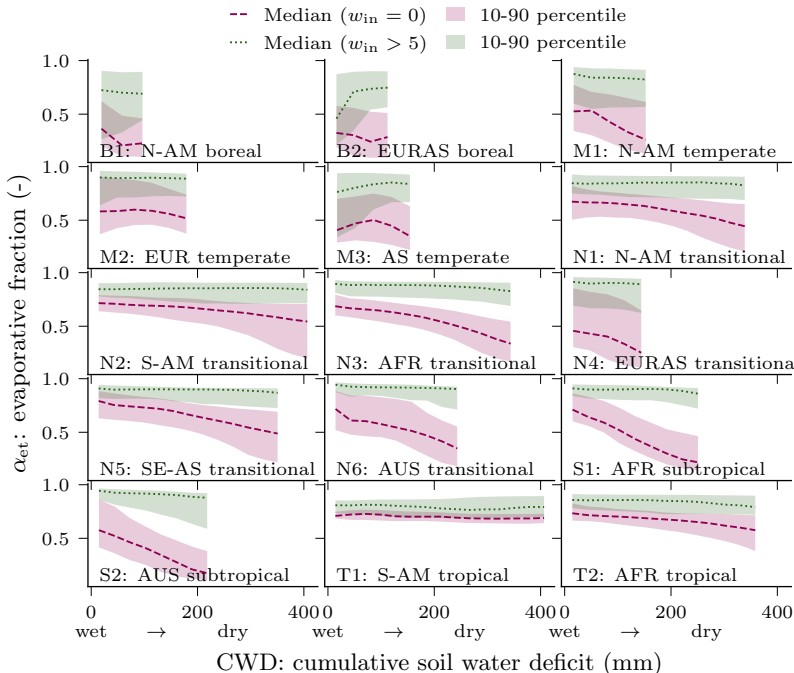

**Figure 7.** Relationship between evaporative fraction ($\alpha_{et}$) and cumulative soil water deficit (CWD) for different hydroclimatic regions. The lines shows the respective percentiles of the spatio-temporal conditional distribution $P(\alpha_{et} \mid C \in B_i)$, where $C$ is the cumulative soil water deficit on the x-axis discretized into $N = 10$ bins $B = \big\{[0, 40), \ldots, [360, 400)\big\}_{i=1,\ldots,N}$. The lines represent the median, and the 10 to 90th percentile is displayed as shaded area. The red colors depict conditions without water input, $P(\alpha_{et} \mid C \in B_i, w_{in} = 0)$, i.e., no precipitation or snowmelt, and green colors represent high water input larger than 5 mm, $P(\alpha_{et} \mid C \in B_i, w_{in} > 5)$. Note that the CWD minimum was subtracted per grid cell. To exclude cells with a low CWD variability, only the cells in the top 60 percent maximum CWD were used. The CWD dynamics correspond to negative soil moisture, i.e., a larger CWD implies dryer soils. The plots are based on global daily cell time steps from 2009 to 2014.

In most hydroclimatic regions, $\alpha_{et}$ showed a negative relationship to CWD under dry conditions (magenta lines in Fig. 7), and no relationship in presence of precipitation or snowmelt (green lines in Fig. 7). The high latitude and tropical regions showed a less clear relationship and less variation in CWD in general. In all regions, $\alpha_{et}$ was close to 1 with large water input

($w_{in} > 5$ mm). In arid (S1-2) and semiarid (N1-5) climates, $\alpha_{et}$ exhibits a large range with steep gradients given low water input ($w_{in} = 0$ mm), decreasing with larger CWD (dryer soil). The 10–90th percentile spread is large in most cases, which indicates that the relationship is modeled with a large spatio-temporal variability.

The H2M shows a large water balance surplus of 12.9 and 21.4 mm yr$^{-1}$, respectively, depending on the forcing dataset used (Tab. 4). The values are robust across cross-validation runs. The largest surplus occurs with the GPCP precipitation product,

which is 9 mm yr$^{-1}$ larger than WFDEI. The GHMs all show a lower ET and a larger Q trend than H2M.

**Table 4.** Global yearly evapotranspiration (ET), grid cell runoff (Q), precipitation (Precip.), and storage change ($\Delta$ Storage) over the period from 2003 to 2012 for the hybrid hydrological model (H2M) and a set of physically-based global hydrological models (GHMs). The H2M was forced with the GPCP precipitation product ("H2M") and the WFDEI data ("H2M (WFDEI)") independently. The latter dataset is also used by the GHMs. The values for H2M and H2M (WFDEI) represent the mean $\pm$ the standard deviation across all cross-validation runs. Values from the common land-mask of all models were considered.

| Model | ET (mm yr$^{-1}$) | | Q (mm yr$^{-1}$) | | Precip.[*] (mm yr$^{-1}$) | $\Delta$ Storage (mm yr$^{-1}$) |
|---|---|---|---|---|---|---|
| H2M | 564 | $\pm$ 6.7 | 274 | $\pm$ 6.5 | 860 | 21.4 $\pm$ 1.1 |
| H2M (WFDEI) | 553 | $\pm$ 6.0 | 285 | $\pm$ 6.5 | 851 | 12.9 $\pm$ 1.0 |
| W3RA | 515 | | 332 | | 851 | 2.5 |
| LISFLOOD | 468 | | 397 | | 851 | -14.3 |
| SURFEX-TRIP | 552 | | 296 | | 851 | 2.3 |
| PCR-GLOBWB | 504 | | 348 | | 851 | -1.3 |

[*]GPCP for H2M, else WFDEI.

The global parameters ($\beta$) were both estimated robustly, with a mean baseflow constant $\beta_{gw} = 0.008$ and a mean snow undercatch correction constant $\beta_{snow} = 0.77$ and a relative standard deviation of 6 % and 2 % across the 15 cross-validation runs, respectively.

### 3.4 Terrestrial water storage composition

In this section, we show the TWS partitioning into snow, soil moisture, and groundwater variations as simulated by H2M and compare it with the corresponding partitioning from the GHMs.

The spatial patterns of the TWS partitioning vary strongly among the models (Fig. 8). Some patterns are consistent, though: The TWS seasonality (Fig. 8, top) is dominated by SWE in the high latitudes in all model simulations. Furthermore, all models tend to attribute the TWS variability to soil moisture in hot arid and semiarid climates. In other regions, the models diverge

substantially. Both W3RA and PCR-GLOBWB attribute stronger groundwater contributions in most tropical and mild climates, while LISFLOOD and SURFEX-TRIP do not show much variation outside cold, semiarid, and arid regions. In H2M, only the humid Amazon and Southeastern Asia show a distinct contribution from groundwater. For the TWS$_{IAV}$ decomposition (Fig. 8, bottom), we see a rough agreement between the H2M, LISFLOOD, W3RA, and PCR-GLOBWB model in North America, Europa, and northern and central Asia. The latter two again show a stronger groundwater contribution, which extends to

southern tropical and mild climates. The largest difference between H2M and the GHMs is the low H2M contribution of groundwater to TWS$_{IAV}$ in Africa, which could also be seen in the TWS$_{MSC}$ decomposition (Fig. 8, top).

Not only the spatial patterns of the TWS partitioning show large variations. Figure 9 illustrates the differences in amplitude and timing for the global time series and their decomposition into MSC and IAV. For the seasonal TWS signal, the amplitudes are qualitatively similar, and the main contribution comes from the snow. H2M, SURFEX-TRIP, and PCR-GLOBWB show a

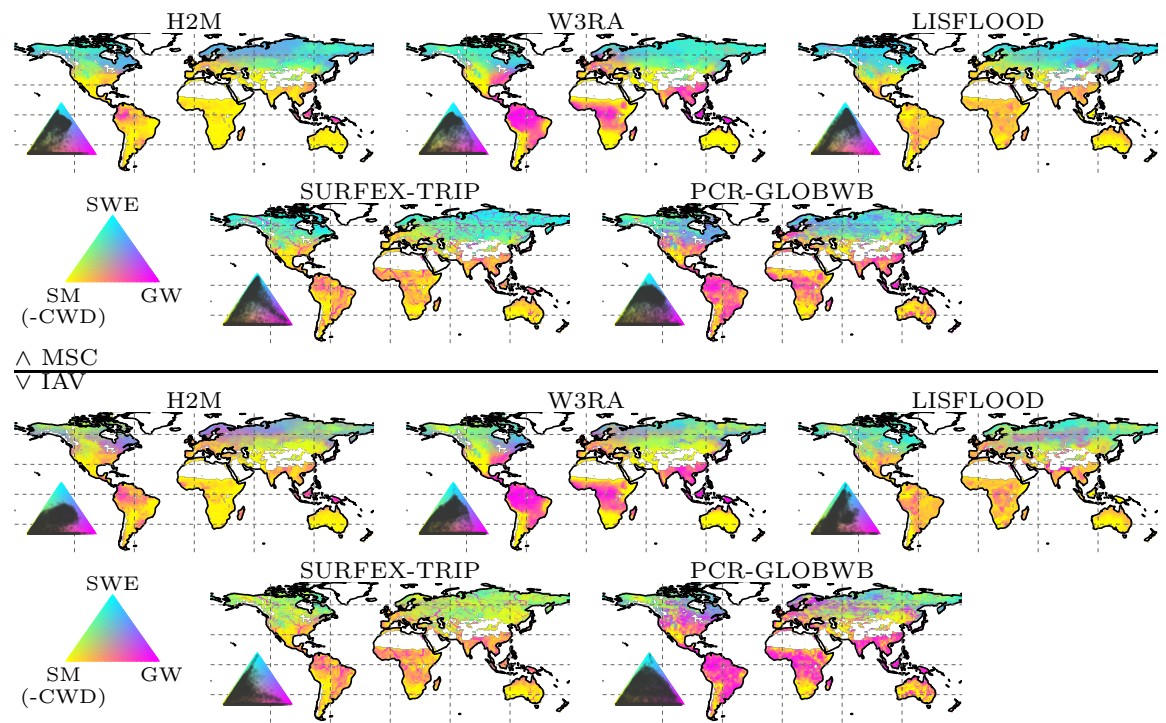

**Figure 8.** Terrestrial water storage (TWS) variation partitioning into soil moisture (SM, corresponding to negative modeled cumulative water deficit, CWD), groundwater (GW), and snow water equivalent (SWE) by the hybrid hydrological model (H2M) and a set of process-based global hydrological models (GHMs). The top panels show the partitioning of the mean seasonal cycle (MSC), the bottom the interannual variability (IAV). The map colors correspond to the mixture of the contributions of the three variables, the inset ternary plots reflect the density of the map points projected onto the components. The contribution is calculated as the sum of the bias-removed absolute deviance of a component from the mean, divided by the contribution of all components. Note that surface storage is included in the groundwater component for the models SURFEX-TRIP and PCR-GLOBWB. The decomposition is done based on the years 2003 to 2012.

soil moisture slightly delayed to the snow seasonality, and the groundwater peak setting in in the late northern spring. W3RA shows very similar soil moisture and groundwater curves, being slightly delayed to the snow seasonality, and LISFLOOD simulates groundwater and soil moisture in alternating cycles with only little variability. The IAV timings of the components are more consistent, but the amplitudes differ significantly across the models. The H2M attributes most $TWS_{IAV}$ to variations in soil moisture, while groundwater dominates the signal for PCR-GLOBWB. Note that the groundwater component also
includes the surface water storage for the latter. Also, SURFEX-TRIP and PCR-GLOBWB both show a large global negative IAV anomaly from 2005 to 2006 and a positive one from 2008 to 2010, which are not observed by GRACE.

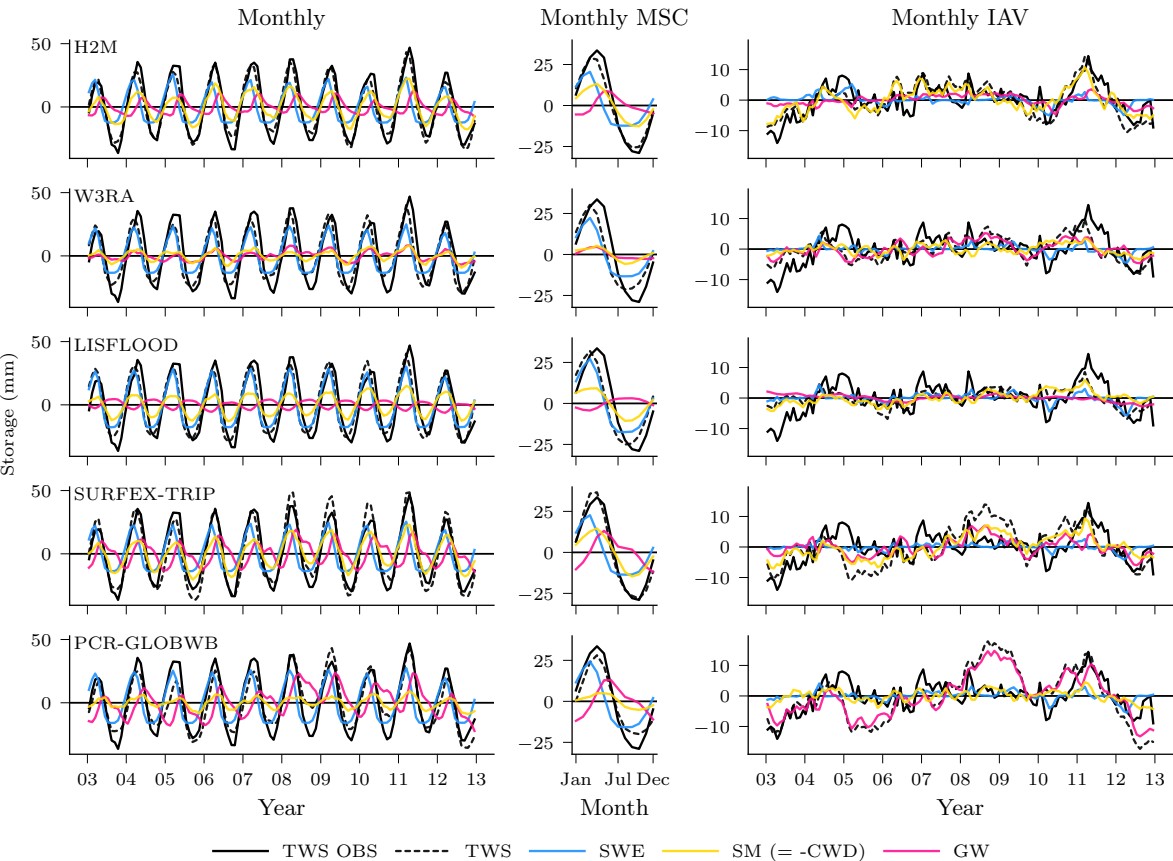

**Figure 9.** Global variability of the terrestrial water storage (TWS) and the components snow water equivalent (SWE), soil moisture (SM), and groundwater (GW) for the hybrid hydrological model (H2M) and the process-based global hydrological models (rows). Note that SM corresponds to negative modeled cumulative water deficit (CWD) in H2M. For reference, the TWS observations are shown (TWS OBS). The monthly signal (left) and its decomposition into the mean seasonal cycle (MSC, center) and the interannual variability (IAV, right) are arranged in columns. The time series represent the global signal, i.e., the data were aggregated using the cell size weighted average per time step, only cell time steps present in all model simulations were used. The y-scale is consistent in columns but varies across the signal components. The training and test period is shown for the complete years 2003 to 2012. Note that surface storage is included in the groundwater component for the models SURFEX-TRIP and PCR-GLOBWB.

## 4   Discussion

In this section, we briefly discuss the model performance and then assess the plausibility of a set of hydrological responses in H2M. We discuss the machine-learned relationship between CWD and runoff generating processes, followed by an analysis of the CWD-$\alpha_{et}$ (evaporative fraction) relationship. Then, we shed some light on the contrast of TWS composition between H2M and GHM simulations. Finally, we discuss general challenges and opportunities of the hybrid approach.

## 4.1 Model performance

The H2M simulations have a good agreement with the TWS and SWE observations despite the data biases. While some GHMs performed well at the global scale, H2M shows evidences of data-adaptability at the local scale. This can be attributed to the data-driven patterns injected through the neural networks.

The TWS seasonality was reproduced well by H2M, except for extremely arid climates, with a low signal-to-noise ratio in observation, resulting in poor NSE values but also small RMSE and decent Pearson's correlation. The largest errors occur in humid regions with a stark TWS seasonality and large runoff rates, e.g., the Amazon basin, central Africa, and Southeast Asia (Fig. A1). This may be related to the missing representations of lateral flow or surface water storage variations in general, which can be important TWS contributions in humid environments (Kim et al., 2009; Scanlon et al., 2019), but also to data biases. A near-perfect fit was achieved for the globally averaged SWE seasonality (Fig. 5) while the local performance varied strongly across regions with the poorest performance in extremely cold tundra (Fig. B2). The $SWE_{IAV}$ is highly sensitive to the precipitation forcing data, which is highlighted by substantially better agreement with GlobSnow when H2M was forced with the WFDEI dataset (Fig. D1 in Appendix D).

In the hybrid modeling framework, the quality of the observational constraints is a major source of uncertainty. The data used in this study have well-documented deficiencies: The precipitation product, for example, shows large uncertainties in Africa due to limitations in density and quality of measurement sites (Sylla et al., 2013) and exhibits biases in snowfall estimates in the Northern Hemisphere due to over-correction of snowfall under catch (Behrangi et al., 2016; Panahi and Behrangi, 2019). The GlobSnow SWE saturates above $120\,\mathrm{mm}$ and underestimates the interannual variability (Luojus et al., 2010). TWS quality is generally difficult to quantify as an equivalent ground-based measurement does not exist, and its complex preprocessing has known impacts on the data quality (Scanlon et al., 2016). The machine learning–based constraints of Q and ET are not directly observed and thus, they are expected to have considerable global and regional uncertainties and biases (Ghiggi et al., 2019; Jung et al., 2020). This could lead to inconsistencies in the water balance~(Trautmann et al., 2021). However, the multi-objective optimization may dampen negative effects of biases, as the model can trade off the different constraints.

## 4.2 Model interpretability

In this section, we assess the model interpretability, i.e., the plausibility of the hydrological responses that emerge from the machine learned coefficients which have not been prescribed a-priori. We discuss the partitioning of water fluxes and their dependence on antecedent soil moisture condition and then evaluate the partitioning of water storage contributing to TWS dynamics.

### 4.2.1 Hydrological responses

The H2M learned hydrological responses to soil moisture states that are consistent with the hydrological understanding, and the learned coefficients are estimated robustly across cross-validation runs. The fact that these patterns are an emerging be-

havior constrained by a basic physical constraint of mass balance, i.e., the relationships were not explicitly predefined, is an encouraging finding that justifies the usage and further investigation of the hybrid approach, in general.

The partitioning of incoming water into surface runoff and recharge of the soil and groundwater shows a clear non-linear response to soil dryness (Fig. 6). The fraction of surface runoff ($\alpha_{surf}$) decreases rapidly with increasing dryness while soil recharge ($\alpha_{soil}$) increases correspondingly. Groundwater recharge occurs under wet conditions and approaches zero with increasing soil dryness. This runoff generating process response to soil moisture qualitatively matches the expected behavior implemented in GHMs (Bergström, 1995).

The H2M predicts a large spatial-temporal variability of the soil moisture dependent runoff-recharge partitioning as indicated by different percentiles in Fig. 6. For example, under moist conditions, more than 50 % of water input (blue lines in Fig. 6) or hardly anything (yellow lines) can be directed to surface runoff. Such large variability in the response can be expected due to large variations of topography, soil, and vegetation properties that control the infiltration-runoff response. The H2M approach, therefore, appears to offer perspectives in capturing the large natural variability of the effective runoff generating process 575 response. Note that these processes have been challenging to parameterize in traditional GHMs (Döll and Flörke, 2005; Beck et al., 2016, 2017; Koirala et al., 2017), and thus, the hybrid approach can fill in critical process gaps and long-standing physical modeling challenges.

    The learned relationship between evaporative fraction ($\alpha_{et}$) and soil dryness (Fig. 7) is generally consistent with the "demand-supply" framework for evapotranspiration (Budyko, 1974). Under wet conditions, ET scales with atmospheric demand repre-580 sented by net radiation, while evaporative fraction declines with increasing dryness which is most clearly seen in the semi-arid regions of Australia and Africa. The learned relationship between $\alpha_{et}$ and soil moisture response functions appear to be rather gradual as opposed to an idealized piece-wise function with a clear soil moisture threshold that is still frequently employed in process models (Seneviratne et al., 2010; Schwingshackl et al., 2017). However, an about constant potential evaporative fraction was predicted when there was substantial rain (or snowmelt), independent of the soil moisture state (green lines in Fig. 7). 585 This shows that the model implicitly accounts for wetting of the top soil layers, which alleviates water stress even though it represents soil moisture (expressed as negative CWD) as a single bucket. The specific response of evaporative fraction predicted by H2M varies substantially between regions and within regions indicated by the shading in Fig. 7. Vegetation storage capacity has long been identified as a key uncertainty in process models in controlling soil moisture stress responses (Ichii et al., 2009). Our approach in H2M avoids such explicit parameterizations of relatively less understood physical processes, and 590 its effectiveness is supported by better performance of H2M in simulating TWS variations in tropical and subtropical regions compared to GHMs (Sect. 3.2) despite its simple overall structure.

### 4.2.2   Terrestrial water storage composition

As reported previously (Andrew et al., 2017) and as presented here, the attribution of TWS variations is an outstanding challenge in global hydrology. The fact that all models disagree largely in respect to the decomposition was the main motivation 595 to use an alternative, data-driven hybrid approach. The decomposition patterns simulated by H2M are reasonable, although the ground truth for a quantitative assertion is missing. The H2M simulations agree with the GHM especially in regions where

the decomposition is well constrained, which is an encouraging finding. In the tropical and semi-arid to arid regions, the decomposition is less clear. Here, all models disagree, although the larger soil moisture variations versus smaller groundwater variation is a unique feature of the H2M simulations. This may indicate that H2M is underconstrained in these regions. Or, the differences could result from a more accurate representation of the involved processes due to the local adaptivity of H2M. Most likely, it is a combination of both.

The dominant contribution of the SWE to seasonal cycle of TWS in the high latitudes (Fig. 8 & 9), but a lower contribution to the interannual variability is consistent across models, and also has been previously reported (e.g., Rangelova et al., 2007; Trautmann et al., 2018). It should be noted that the $SWE_{IAV}$ was reproduced poorly by all models, reflecting large uncertainties in the input precipitation and SWE observations. Despite regional differences, the models also consistently attribute most of the TWS seasonal and interannual variability to soil moisture in arid and semi-arid regions (Fig. 8). The dominance of soil moisture is plausible in these regions, as the potential evapotranspiration is high and precipitation is low and infrequent or strongly seasonal (Nicholson, 2011). Given the absence of secondary moisture sources such as lateral flow and a lack of deep-rooted plants, most of the storage variations occur within a shallow soil depth (Grayson et al., 2006).

In other regions, the partitioning between groundwater and soil moisture variability is less clear. On both the seasonal and interannual scales, groundwater contributions to TWS correlate with humidity at the global scale (c.f., Feddema, 2005). In the boreal humid regions of northwestern North America, Scandinavia, and northwestern Russia, as well as the northeastern Asian coast, the groundwater contribution to TWS is larger than that of soil moisture. Here, groundwater recharge is concentrated in spring with large snowmelt (Fig. 9) co-occurring with low evaporative demand due to low temperatures, irradiation, and vegetation productivity, which results in a large water surplus (Jasechko et al., 2014). The boreal regions with stronger soil moisture contribution are the ones affected by permafrost, where most of the vertical movement is limited to the thawed top soil and horizontal baseflow is usually lower than in non-permafrost soils (Bui et al., 2020). Thus, the patterns diagnosed by H2M are plausible. It must be noted, however, that significant drainage of the surplus water happens via river flows and lateral transport, which are not represented in H2M.

The large groundwater contribution on both seasonal and interannual scales in humid regions has been diagnosed by all models. In the tropics, the largest difference between H2M and the GHMs is the larger soil moisture contribution in the African rainforest simulated by H2M. The lower groundwater variability is—to a certain extent—reasonable, as the central Amazon and Southeast Asia rainforests are the most humid regions globally with the largest annual precipitation (Zelazowski et al., 2011) and a shallow plant rooting depth, while the African rainforest is relatively drier and has deeper plant roots (Yang et al., 2016; Fan et al., 2017). However, the soil moisture variability is only marginally larger in H2M, while it is mainly the low groundwater amplitude that makes the difference (Fig. B3 in Appendix B).

In the arid-to-wet transition regions of Africa, H2M diagnoses only marginal groundwater variability compared to larger amplitudes in the GHMs. The H2M resolves the water balance mainly using soil moisture variations, i.e., through soil recharge and evapotranspiration, while the soil overflow was negligible. While the patterns found by H2M are within those of GHMs in most regions, the notable strong soil moisture contribution in tropical savanna and humid subtropical climates is unique in H2M.

GHMs require a large number of parameters that are either empirically derived or based on remote sensing or statistical datasets, e.g., plant functional types, root zone depth, soil texture maps, or soil thermal and hydraulic properties. Often, the said parameters are uncertain and may not necessarily represent a process at spatial scale of GHMs (scale mismatch) or within grid or catchment variabilities (sub-grid to local heterogeneity). Thus, simple heuristics have been used to parameterize hydrological processes, which can, in reality, be of high complexity (Beck et al., 2016). It has been suggested that GHMs underestimate the land water storage capacity in general and that especially the variability in deeper soil is too low (Zeng et al., 2008). In addition, the link between deeper soil layers and plant transpiration through root water uptake is often not represented adequately in GHMs (Jackson et al., 2000), although such effects have been found to play an important role in below-surface water variability (e.g., Kleidon and Heimann, 2000; Koirala et al., 2017). Compared to the GHMs, H2M provides a novel avenue on which storage variations are less bound by presumably ad-hoc prescription of the size of soil and other storages. The diagnosed patterns of soil and groundwater variations, therefore, emerge from observation-based variations of water storage and fluxes. The H2M approach that also implicitly learns layering of the soil, thus, can be used to address uncertainties in the moisture storage capacities (Zeng et al., 2008; Scanlon et al., 2019) and plant rooting depth (Yang et al., 2016) used in GHMs, that are likely to have a strong influence on the TWS partitioning.

The smaller groundwater contribution in H2M is also potentially related to the missing mechanisms of capillary rise and root water uptake from the groundwater. Thus, the cumulative water deficit dynamics implicitly represent all the below-ground water that will be returned to the atmosphere by root water uptake and transpiration at some point. As a possible consequence, H2M diagnoses larger soil moisture in transitional and especially in the subtropical regions, but more evidently, smaller groundwater variability.

Finally, the missing (explicit) representation of surface water and river storage may cause biases in H2M simulations. Surface storage has been found to contribute significantly to the TWS variations (Güntner et al., 2007; Scanlon et al., 2019) and a proper representation thereof is desirable. Furthermore, lateral water influx across a cell via rivers is not represented and may have a significant impact on the TWS composition (Kim et al., 2009).

### 4.3 Challenges and opportunities

The data-driven character of the H2M offers a set of opportunities but is accompanied by challenges. The H2M makes use of observational data streams that are not typically used in GHMs. However, to retain interpretability of the predicted coefficients, the model structure must be kept simple; the model flexibility needs to be compensated with a simple causal model structure. Still, the H2M offers a great opportunity to study the hydrological cycle from a different viewpoint that is strongly footed on the observation-based datasets, which are growing in availability at an unprecedented rate in the era of Earth observation.

The hydrological pathways in H2M are rather simple compared to GHMs, but the model still expresses a high data-adaptivity as demonstrated. While GHMs usually represent a wide range of hydrological sub-processes (e.g., infiltration, preferential flow, topographical runoff-runon), the hybrid model integrates them to a few response functions and the model complexity and interactions within is, so to speak, outsourced to the neural network. Still, missing representations of storage components (e.g., surface storage) and hydrological pathways (e.g., streamflows) limit the model flexibility and can, to a certain extent,

corrupt the other latent variables as the model tries to accommodate for missing processes. Thus, the estimated coefficients in the current H2M implementation should be treated with some skepticism. At the same time, the relaxation of assumptions can be seen as an opportunity, as the prior knowledge used in GHMs may be wrong or incomplete. The impact of trading prior knowledge and model complexity with more flexibility and data-drivenness on model uncertainties is a key aspect that needs

further investigation.

As the model behavior emerges largely from the observational data constraints, the hybrid approach constitutes a novel technique for studying TWS variations. While purely data-driven approaches (see Andrew et al. (2017) for an overview) are generally useful as they provide insights independent from GHMs, they are based on strong qualitative assumptions (e.g., the temporal characteristics of the components at different depths) and do not allow to incorporate physical knowledge, principles,

and constraints. GHMs themselves largely rely on prior knowledge, which may be false or incomplete, and the model parameterization is usually not resolved regionally, resulting in model uncertainties (Beck et al., 2016) which are eventually expressed in the disagreement among model simulations. The hybrid model can be seen as a compromise between the purely data-driven and the physically-based approaches, as physical principles (e.g., mass conservation) are respected, but qualitative assumptions on the processes are still used.

Global hydrological models are often used for different tasks such as the assessment of the water cycle at past and present, predictions for the future, for evaluating implications of, e.g., land use changes by scenarios, and to gain process-understanding. In principle and technically, a global hybrid hydrological model can be applied for the same tasks while related simulations need to be interpreted with care. The strongest use case of H2M is the assessment of recent variations of the water cycle since it can act as a physically consistent yet data-adaptive bridge between heterogeneous global data streams and complements

traditional data assimilation approaches. Interpreting predictions too far into the past or future can be risky when factors that are not represented physically play a role that had little impact during the training period (e.g., permafrost melting, CO2 fertilization of water use efficiency). Likewise, scenarios of, for example, different land use could make sense to conduct if the conditions represented by the scenarios have been represented during learning in some way while there always remains the danger that learned relationships by the neural network are just statistical associations rather than causal relationships ("shortcut

learning", Geirhos et al., 2020). As we could show, gaining process understanding from the hybrid model can be feasible as the spatially and temporally varying coefficients learned by the neural network are plausible and partly very interesting. However, such uncovered patterns may rather represent hypotheses that should be tested with complementary approaches like physical process modeling, direct observations, or experiments.

Improving the model through a better representation of the process complexity is an obvious next step. Several processes

were not explicitly represented, such as overland flow, soil moisture recharge from the groundwater through capillary rise, or snow sublimation. The under-complex representation of certain processes leads to biases and uncertainties. For example, estimating the baseflow parameterization on cell-level could improve the representative power of the model, as has been shown by Beck et al. (2013). This is, however, challenging as an increasingly complex model needs to be complemented by additional data constraints or better physical processes in order to avoid parameter equifinality issues that lead to the same or similar

model responses across a large range of parameter values. It is well possible that the decomposition into CWD and GW is not

properly constrained under some circumstances, e.g., in ecosystems that are not water limited. Here, either the groundwater or the soil moisture may be restored as needed (due to frequent precipitation) to match the observation of terrestrial water storage. More research is needed to address these problems. In particular, a complementary development of application-based models as presented in this study, and smaller-scale, better constrained exercises to advance hybrid modeling can be a viable alternative.

Closely related to equifinality is the quantification of model (epistemic) and data (aleatoric) uncertainties. A proper representation of model uncertainties would enable a direct identification of equifinality and allow a targeted model development for uncertain processes. The implementation of such a mechanism could be built into the neural network, e.g., by using Bayesian deep learning (Wang and Yeung, 2020) or deep generative models (Goodfellow et al., 2016). Explicit consideration of data uncertainty will also be beneficial, either to propagate forcing data uncertainties through the model or to model the uncertainties of the observational constraint variables, which is not always provided. Data assimilation is a framework that allows representing such uncertainties (Reichle, 2008) and can even be extended to incorporate model parameter estimation (Moradkhani et al., 2005), i.e., learning physical processes as in the hybrid approach presented here. In contrast to data assimilation that often targets improving prediction skills, the goal of hybrid modeling is to develop a generalizable model, which can be applied beyond the specific forecasting task in data assimilation. Nevertheless, non-parametric machine learning approaches can also be included into data assimilation as discussed in Geer (2021).

The rapid development of novel products opens interesting opportunities, like a daily TWS product (Kvas et al., 2019) can help to better constrain sub-monthly water processes. Furthermore, the upcoming Surface Water and Ocean Topography (SWOT) mission, which is targeted at observing surface water storage variations (Biancamaria et al., 2016), could be useful to solve current shortcomings of the H2M. In addition, parameters estimated by other approaches, such as the upscaled baseflow index (Beck et al., 2013), offer interesting independent constraints that allow adding further complexity to the model without increasing the uncertainty.

Finally, incorporating lateral interactions and flow between grid cells (e.g., large-scale groundwater flow, river routing) are outstanding but relevant challenges, as the paradigm of optimizing neural networks with randomized samples that are independent will likely not be sufficient in modeling connections and interactions between regions. Such endeavors would also allow for bringing in established global datasets of river discharge measurements such as provided by the Global Runoff Data Centre (GRDC, Fekete et al., 1999).

## 5 Conclusions

The present study demonstrates the strengths of combining machine learning and physical process understanding for global hydrological modeling. The main conclusions of this study are:

1. The hybrid model is capable of obtaining similar performance to physically-based models at global level but achieved better local adaptivity. This highlights the strengths of the hybrid approach, which can replace complex physical processes, integrate different datasets, and is highly data-adaptive due to the model parameterization by a neural network.

2. The model simulations were plausible and followed basic hydrological principles. This is partially due to the physical constraints, which force the model into physical consistency (e.g., conservation of mass), but is also emerging from the multiple data constraints.

3. The hybrid model partitioning of the terrestrial water storage into its components yielded plausible and interesting patterns. The agreement of the decomposition is generally high in regions where the physically-based models are more consistent (e.g., temperate, semi-arid, and arid regions), but generally the hybrid model shows a larger contribution by soil moisture.

4. Key opportunities and challenges in hybrid modeling to be addressed in the future are identification of equifinality, quantification of uncertainties, integration of multi-resolution datasets, and representation of cell-neighborhood effects, such as lateral fluxes.

Hybrid modeling has the potential to advance the Earth sciences by providing an alternative perspective to knowledge-driven approaches. The data-adaptivity can reveal weaknesses and strengths of process-based models and provide important insights for water cycle attribution and diagnostics. The findings and methods of this study can be generalized to other spheres and scales across the Earth system, as long as sufficient data and process knowledge are available.

*Code and data availability.* The H2M and its training are implemented in *PyTorch 1.5* (Paszke et al., 2017), an open-source deep learning framework for the *Python* programming language. The simulated hydrological data and the code are available here: https://dx.doi.org/10. 17617/3.65. The code is also available on github: https://github.com/bask0/h2m. Note that we cannot share the data used as model input, but all datasets are referenced in the manuscript.

## Appendix A:  Spatial model performance

Overall, high NSE of TWS$_{MSC}$ is achieved in most regions (Fig. A1). Low TWS$_{NSE}$ hotspots are primarily found in some arid regions with little overall TWS variability, e.g., the Namib Desert in southern Africa or the Gobi Desert in eastern Asia. In terms of the RMSE, regions with larger variations in TWS dominate with the largest MSC error in the Amazon and less expressed in southeastern Asia. The correlation ($r$) was constantly well above 0.5 for TWS$_{MSC}$ except for the Gobi Desert, where the TWS variations are minimal. The TWS$_{IAV}$ was also reproduced well in terms of $r$.

The SWE$_{MSC}$ is reproduced well in terms of NSE and $r$, while NSE for SWE$_{IAV}$ is low especially in tundra regions (Fig. A1). The RMSE is also larger in high latitudes but more concentrated in regions with large seasonal amplitudes.

The average patterns of states (TWS and SWE) and fluxes (ET and Q) were reproduced well in general (Fig. A2). The model underestimates the variability of TWS in central Amazon, West Africa, and India. These patterns align well with the occurrence of large rivers (e.g., Amazon, Ganges, Mississippi, Niger, or Yenisei) and may be caused by missing representation of river routing. The SWE is overestimated in the extremely cold regions of North America and Northeast Asia, and underestimated

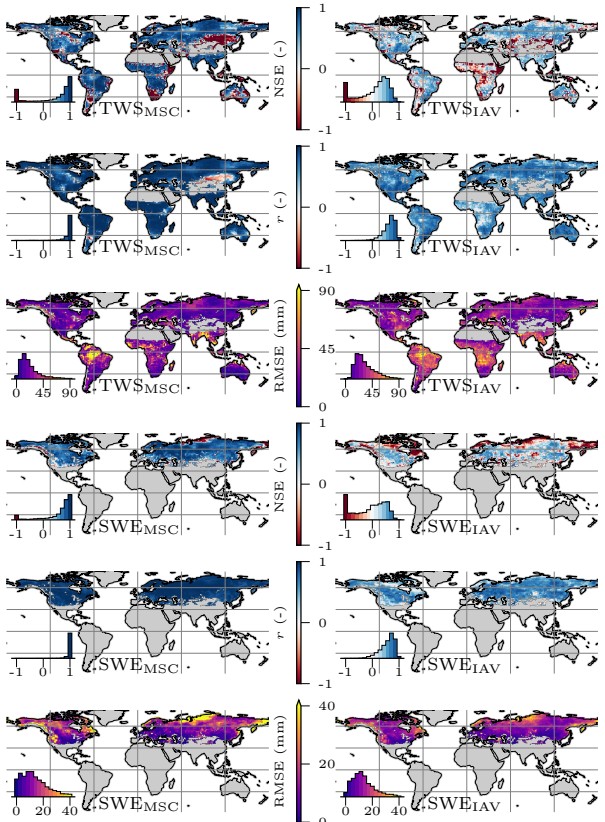

**Figure A1.** Local model performance for terrestrial water storage (TWS) and snow water equivalent (SWE) on the mean seasonal cycle (MSC) and the interannual variability (IAV) within the test period (2009 to 2014). The Nash–Sutcliffe model efficiency (NSE), Pearson correlation ($r$) and Root Mean Square Error (RMSE) are shown. The inset plots show the cell area–weighted histogram of the map values.

in Tundra regions. Average Q is largely underestimated in Central Africa, and slightly overestimated in northwestern Eurasia, central Amazon, and coastal regions of Australia and East Asia. ET, finally, is underestimated by the model, prominently in most of Subsaharan Africa and East Brazil, while no major biases are present in other regions.

## Appendix B:  Regional comparison of simulated time series

On regional scale, most models reproduced the $TWS_{MSC}$ well ($e_{NSE} > 0.5$), while the $TWS_{IAV}$ performance varied ($e_{NSE} < 0.5$) (Fig. B1). The variation between models was larger in terms of IAV, especially in transitional and tropical zones. Especially the $TWS_{IAV}$ seems to be reproduced poorly in certain regions by all models, e.g., temperate Asia (M3), transitional Africa (N3), Eurasia (N4), Southeast Asia (N5). In the high latitudes, we observe a phase difference of the simulated TWS compared to the observations for all models except the PCR-GLOBWB.

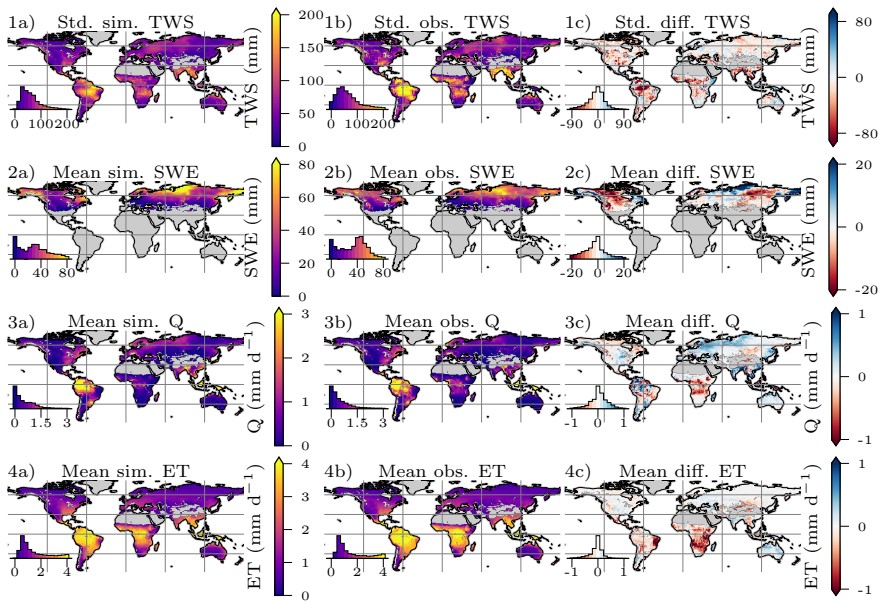

**Figure A2.** Mean a) simulated, b) observed, and c) difference of simulated minus observed (positive means simulated is larger) terrestrial water storage (TWS, 1a–c), snow water equivalent (SWE, 2a–c), total runoff (Q, 3a–c), and evapotranspiration (ET, 4a–c). Note that for the TWS, the standard deviation is shown as the values represent variations around the mean. The inset histograms represent the map value distributions, the mean for the test period (2009 to 2014) is shown.

Most models manage to reproduce the $SWE_{MSC}$ well with an $e_{NSE} > 0.5$, while the $SWE_{IAV}$ performance is more variant and lower in general (Fig. B2). We note a phase difference between the model simulations and observations that is most notable in
the boreal regions, indicating that the models either accumulate too much snow during winter or do not manage to discharge it in spring or both. The phase difference is less expressed in H2M and lowest in PCR-GLOBWB. The $SWE_{IAV}$ varies strongly across different regions. The $SWE_{IAV}$ has strong seasonal variations, with opposite patterns in different regions that cancel each other out on global level. This is evident on the regional anomalies and results in low variability at the global scale. In general, all models reproduce the sign of anomalies better than the amplitudes.
The regional scale seasonal anomalies of simulated soil moisture (corresponding to negative CWD in H2M) and GW show a more detailed picture of the model variabilities (Fig. B3). The global scale SM amplitude of H2M is larger than the one of the GHMs (although close to the SURFEX-TRIP model) while the GW variations are smaller in H2M. The largest discrepancies between H2M and the GHMs are in the North (N1) and South (N2) America transitional, the Australia subtropical (S2), and Africa tropical (T2) regions. However, also the within GHM variation is large in most regions. The model simulations agree
relatively well in the temperate regions (M1-3) as well as in the Africa (N3), Eurasia (N4), and Australia (N6) transitional zones.

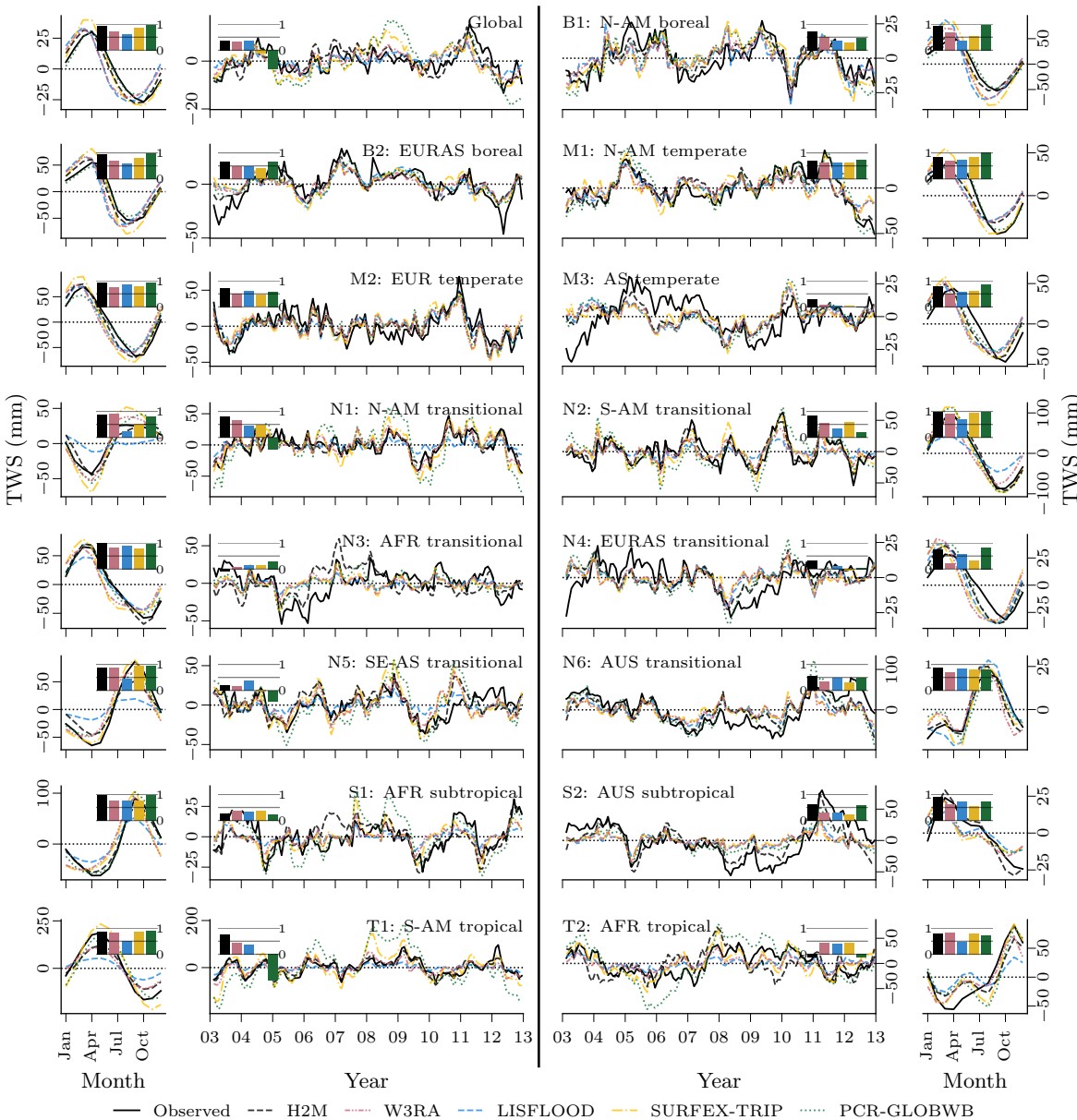

**Figure B1.** Comparison of the hybrid hydrological model (H2M) and a set of process-based global hydrological models (GHMs) of the terrestrial water storage mean seasonal cycle (TWS$_{MSC}$, outer columns) and interannual variability (TWS$_{IAV}$, center columns) in mm for hydro-climatic regions (Fig. 2). The time series were aggregated using the cell size weighted mean across all grid cells in the respective region. The inset axes show the Nash–Sutcliffe model efficiency (NSE) of each model with the same color-coding as the time series. Note that the y-scale differs between plots.

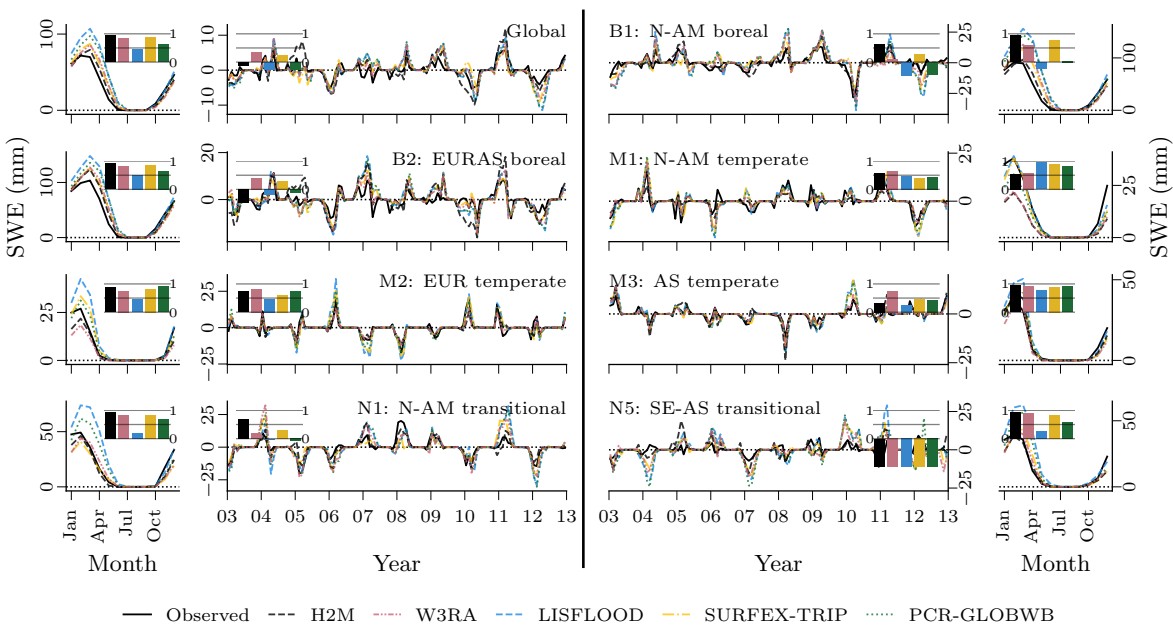

**Figure B2.** Comparison of the hybrid hydrological model (H2M) and a set of process-based global hydrological models (GHMs) of the snow water equivalent mean seasonal cycle (SWE$_{MSC}$, outer columns) and interannual variability (SWE$_{IAV}$, center columns) in $\mathrm{mm}$ for hydro-climatic regions (Fig. 2). The time series were aggregated using the cell size weighted mean across all grid cells in the respective region. The inset axes show the Nash–Sutcliffe model efficiency (NSE) of each model with the same color-coding as the time series. Note that regions without snow dynamics are not included. The y-scale differs between plots.

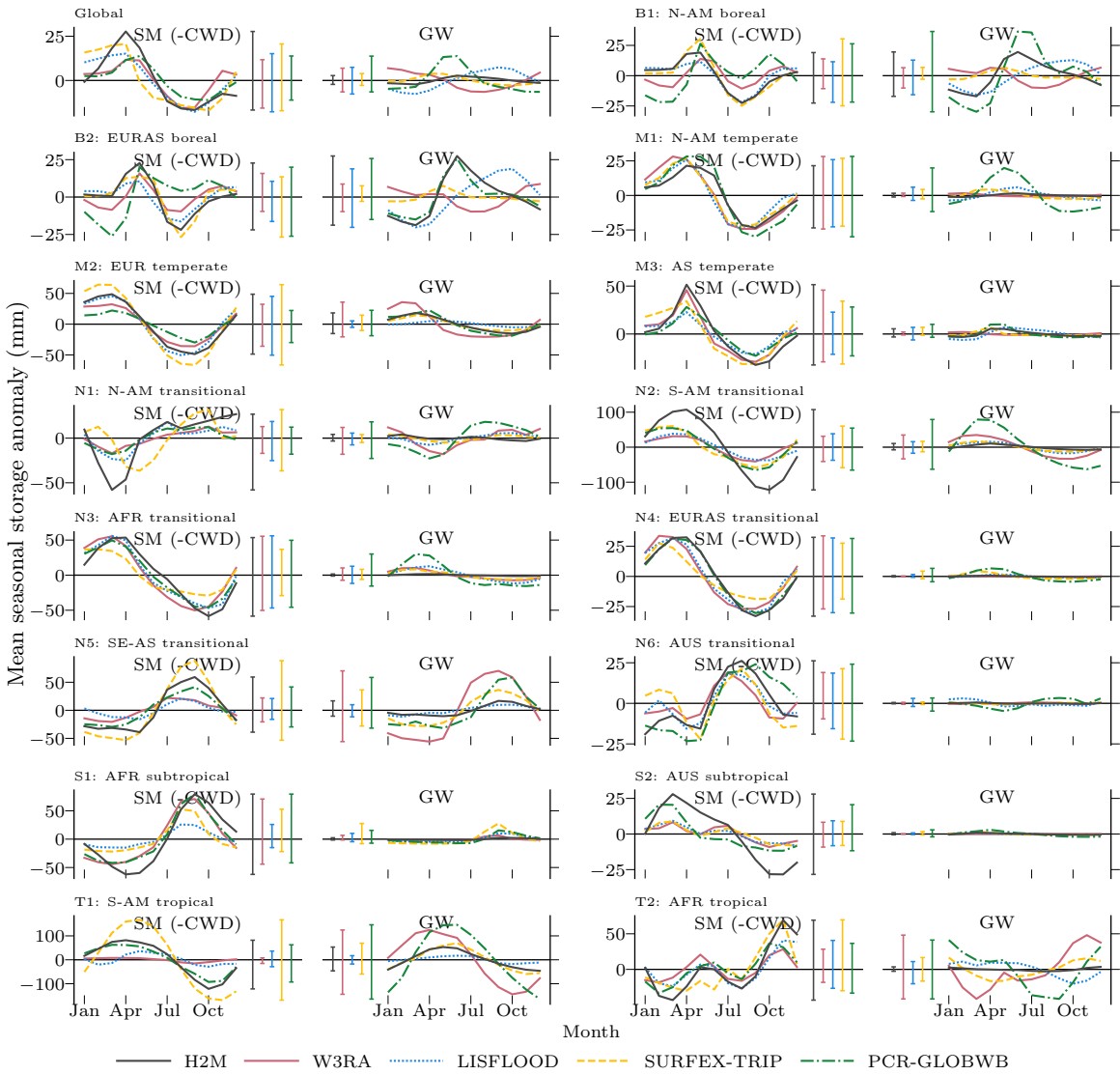

**Figure B3.** Global and regional mean seasonal anomalies of soil moisture (SM) and groundwater (GW) for the hybrid model (H2M) and the process-based global hydrological models. Note that SM corresponds to negative modeled cumulative water deficit (CWD). Ranges from the minimum to the maximum value per model are shown next to the seasonal cycle as vertical lines. The regions are shown in Figure 2. Surface storage is included in the groundwater component for the models SURFEX-TRIP and PCR-GLOBWB. The plots are based on global daily cell time steps from 2009 to 2014. Note that the y-scale is consistent within, but differs across regions.

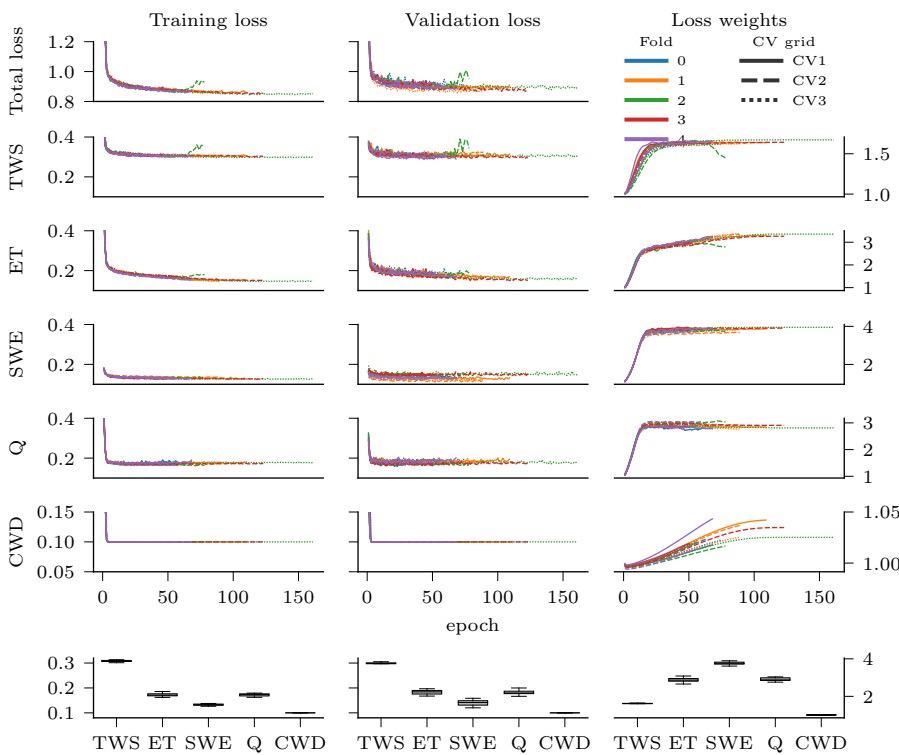

**Figure C1.** Model training process for the cross-validation runs. The left and central columns represent the unweighted total and variable-specific MSE loss. The right column shows how the task weights developed over training time. The x-axis represents the number of iterations through the training set ("epochs"). The bottom row contains the column-wise distribution of the variables losses (or weights) at the end of the model optimization. Note that for the soft constraint on CWD, a bias of 0.1 was added, i.e., 0.1 is the optimum.

## Appendix C: Model optimization

The model optimization within the cross-validation setting is shown in Fig. C1. The learning process was stable in most cases and a smooth model convergence was achieved. Only one run (fold 2, CV2) was unstable as the training collapsed. Due to the early stopping mechanism, however, the model from the best validation loss is restored and used for the test set prediction. The loss and weight ($w = \frac{1}{2\sigma^2}$, where $\sigma$ is the task uncertainty, see Sect. 2.3.3) distributions at optimum across cross-validation runs were stable (bottom row of boxplots in Fig. C1). The generalization loss from the training to the validation loss is minimal, although a slightly larger spread of the validation losses can be observed. The larges generalization error occurred with SWE. Note that the training and validation sets are not only split in space, but also in time. This could indicate that snow dynamics are less stable over time and change due to, for example, a warming climate.

The task weights were stable across cross-validation runs. The weights are difficult to interpret, as they do not directly translate to inverse variable uncertainty (Kendall et al., 2018) but also depend on the variable variance (although the loss is calculated on standardized data). From the boxplots in Fig. C1, we can see that variables with a lower loss are given more

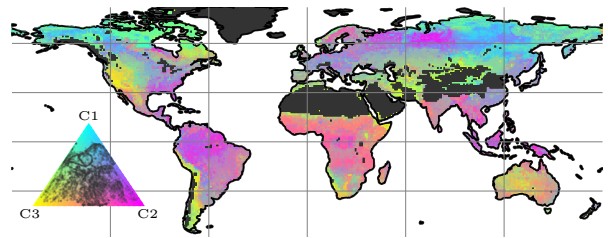

**Figure C2.** The t-distributed stochastic neighbor (t-SNE) reduction to three dimensions (C1-3) of static variable encoding (originally 12 dimensions, $\rho_{enc}$ in Fig. 1) of one cross-validation run. The encoding is a low-level representation of the static inputs, i.e., soil and land-cover properties, learned by a neural network. The inset ternary plots show the distribution of the map values.

weight, except for the CWD loss (a soft constraint that avoids CWD drift in early training), which reaches the optimum at 0.1 relatively quickly. It is possible that the lower weight of TWS is caused by its dependency on the other variables, i.e., if the model tries too hard to improve TWS, other variable losses decrease.

Part of the model tuning involved optimization of the sub-network FCNN[1] (Fig. 1), extracting features from the static variables which are then fed into the recurrent neural network. We visualized the outputs ($\rho_{enc}$ in Fig. 1) of the FCNN[1] to get an impression of the most relevant gradients within the static variables. For visualization, the twelve activations were reduced to three dimensions using t-SNE (Hinton and Roweis, 2002). The resulting map (Fig. C2) reveals patterns that seem very familiar: the components align with patterns of biomass, vegetation type and aridity. Note that the t-SNE algorithm is non-deterministic and can yield vastly different results depending on chosen hyper-parameters. Also, the reduction to three dimensions only reveals the major gradients and does not represent the entire variability.

## Appendix D: Model forcing with WFDEI

To test the impact of the forcing datasets, the model was trained on the WFDEI forcings (Weedon et al., 2014) as used in the eartH2Observe ensemble. The performance (Fig. D1) in respect to TWS was almost identical with slightly larger NSE on the global signal and lower NSE on local level when using WFDEI. The NSE of SWE was larger with WFDEI, especially for the IAV. Due to the similar performance, we conclude that the impact of the forcings is negligible and the results are robust in regards to them.

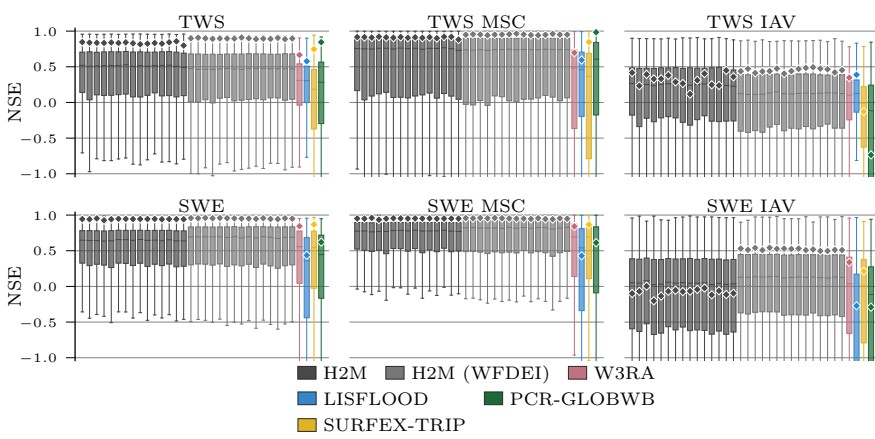

**Figure D1.** Global and local grid cell–level Nash–Sutcliffe model efficiency coefficient (NSE) of the hybrid hydrological model (H2M) and the process-based global hydrological models (GHMs) for the terrestrial water storage (TWS) on top and the snow water equivalent (SWE) at the bottom. The gray bars represent the cross-validation runs using the forcings described in Section 2.1.1 (dark grey, "H2M"), and using the WFDEI forcings as used in the eartH2Observe ensemble (light grey, "H2M (WFDEI)"). The ⬦-markers show the global (spatially averaged signal) model performance, the boxes represent the spatial variability of the local cell-level performance. The y-axis was cut at -1 due to some large negative NSE values. The panels show the model performance in respect to the full-time series, the mean seasonal cycle (MSC), and the interannual variability (IAV). Note that for SWE, only grid cells with at least one day of snow are shown, as the NSE is not defined if the observations are constant zero, which would lead to a comparison of different grid cells. The metrics are calculated from the complete common time range from 2009 to 2012 on monthly time scale. Note that deviations from the numbers reported in Tab. 3 are due to different time ranges.

```
 1:  φ, β, σ ← initialize()                          # Initialize model weights φ, global constants β, task uncertainties σ
 2:  while not converged do
 3:      cells ← sample_cells(gridcells, n)          # sample n gridcells
 4:      m_sim ← meteo[cells]                         # select cells from forcings
 5:      m_spinup ← sample_spinup(m_sim, 5)           # sample 5 random years
 6:      m ← concat(m_spinup, m_sim)                  # concatenate
 7:      ρ ← static[cells]                            # select cells from static
 8:      y ← target[cells]                            # select cells from targets
 9:      c, h ← zeros(100)                            # initialize LSTM hidden states
10:      s ← zeros(3)                                 # initialize physical storages
11:      loss ← 0.0                                   # initialize loss
12:      ρ_enc ← FCNN¹(ρ)                             # compress static encodings
13:      for t ∈ {1, ..., T} do
14:          c, h ← LSTM(c, h, s, m[t], ρ_enc)        # update LSTM states
15:          α ← FCNN²(h)                             # get coefficients
16:          s, f ← hydro(s, m[t], α, β)              # run phys. model, get storages s and fluxes f
17:          ŷ ← collect(s, f)                        # collect target variables
18:          if t ∉ spinup then
19:              loss ← loss + MSE(ŷ, y[t], σ)        # add weighted loss to previous loss
20:          end if
21:      end for
22:      φ, β, σ ← update(φ, β, σ, loss)              # update parameters
23:  end while
```

**Figure E1.** The training loop of the hybrid hydrological model.

## Appendix E: Model pseudo-code

The pseudo-code in Fig. E1 shows the model optimization process.

*Author contributions.* The study was conceptualized by all the authors. BK implemented the model and performed the data analysis in close collaboration with the co-authors. All authors contributed to the manuscript.

*Competing interests.* The authors declare that they have no conflict of interest.

*Acknowledgements.* We thank the International Max Planck Research School for Global Biogeochemical Cycles (IMPRS-gBGC) and the Max Planck Institute for Biogeochemistry for funding and supporting this project. In addition, we thank Uli Weber for data preprocessing and colleagues at the MPI for Biogeochemistry and TU Munich for the stimulating discussions. We are very grateful to the reviewers Lieke Melsen, Derek Karssenberg, the anonymous referees, and Albrecht Weerts, the editor, for helping us improve the manuscript with their comments and suggestions.

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
