# Peer review of "Towards hybrid modeling of the global hydrological cycle"

_Hydrology and Earth System Sciences, 2021_

## Author Comment (AC1)

[revised manuscript text omitted]

---

## Author Comment (AC5)

Kraft et al. (2021), https://doi.org/10.5194/hess-2021-211

**Supplement to the response to the reviewer comments, August 9, 2021**

**Figure A**

[Figure]

**Figure D1.** Global and local grid cell level Nash–Sutcliffe model efficiency coefficient (NSE) of the hybrid hydrological model (H2M) and the process-based global hydrological models (GHMs) for the terrestrial water storage (TWS) on top and the snow water equivalent (SWE) on bottom. The gray bars represent the cross-validation runs using the forcings described in Section 2.1.1 (dark, "H2M"), and using the WFDEI forcings as used for in the eartH2Observe ensemble (light, "H2M (WFDEI)"). The ⋄-markers show the global (spatially averaged per timestep) model performance, the boxes represent the spatial variability of the cell level performance. The panels show the model performance in respect to the full time-series, the mean seasonal cycle (MSC) and the interannual variability (IAV). Note that for SWE, only grid cells with at least one day of snow are shown, as the NSE is not defined if the observations are constant zero, which would lead to a comparison of different grid cells. The y-axis us cut at -1 due to some large negative NSE values. The metrics are calculated from the complete common time-range from 2003 to 2012. Note that deviations from the numbers reported in Tab. 3 are due to different time ranges.

**Figure B**

[Figure]

**Figure 5.** Comparison of the hybrid hydrological model (H2M) and a set of process-based global hydrological models (GHMs) of the terrestrial water storage (TWS), its mean seasonal cycle (TWS$_{MSC}$) and its interannual variability (TWS$_{IAV}$) in mm for the global signal. The time-series were aggregated using the cell size weighted mean across all grid cells. The regional time series are show in Appendix B, Fig. B1.

**Figure C**

[Figure]

**Figure 6.** Comparison of the hybrid hydrological model (H2M) and a set of process-based global hydrological models (GHMs) of the snow water equivalent (SWE), its mean seasonal cycle (SWE$_{MSC}$) and its interannual variability (SWE$_{IAV}$) in mm for the global signal. The time-series were aggregated using the cell size weighted mean across all grid cells. The regional time series are show in Appendix B, Fig. B2.

**Figure D**

[Figure]

Generalization error in terms of RMSE. The boxplots represent the spatial variability of the RMSE per cross-validation fold (colors). For each cross-validation fold, the median RMSE of the respective training set was removed, such that the training median is at zero, and the validation and test set boxplots show the RMSE relative to the training set median.

**Figure E**

[Figure]

Recurrent neural network (RNN) as a loop (left) and unfolded (right). LeCun, Bengio, and G. Hinton, Nature, 2015 / Figure 5.

**Table A**

**Table 4.** Global yearly evapotranspiration (ET), runoff (Q), precipitation (Precip.), and storage change (Δ Storage) over the period from 2003 to 2012. The H2M model was forced with the GPCP precipitation product, the other models with WFDEI. The values for H2M and H2M (WFDEI) represent the mean ± the standard deviation across all cross-validation runs. Values from the common land-mask of all models were considered.

| Model | ET (mm yr$^{-1}$) | Q (mm yr$^{-1}$) | Precip.* (mm yr$^{-1}$) | Δ Storage (mm yr$^{-1}$) |
|---|---|---|---|---|
| H2M | 564 ± 6.7 | 274 ± 6.5 | 860 | 21.4 ± 1.1 |
| H2M (WFDEI) | 553 ± 6.0 | 285 ± 6.5 | 851 | 12.9 ± 1.0 |
| W3RA | 515 | 332 | 851 | 2.5 |
| LISFLOOD | 468 | 397 | 851 | -14.3 |
| SURFEX-TRIP | 552 | 296 | 851 | 2.3 |
| PCR-GLOBWB | 504 | 348 | 851 | -1.3 |

*GPCP for H2M, else WFDEI.

---

## Author Response (AR1)

**Major revisions for the submission *hess-2021-211* "Towards hybrid modeling of the global hydrological cycle"**

Kraft et al.

October 29, 2021

Dear Editor, dear Reviewers

First of all, we would like to express our gratitude for the time and effort that all reviewers and the editor put in to the critical evaluation of study and manuscript. The comments and suggestions were very useful to improve the manuscript and to have a clearer presentation.

It was obvious from many suggestions and comments of all reviewers that the manuscript presentation was missing details of the model structure and training that reflected imbalance of the information content. In response, we have gone through the whole manuscript for improving conciseness, added missing details, reorganized sections, and moved less relevant information to the appendix. We believe the changes make the main manuscript much more streamlined and clear, while the extended appendix still maintains the comprehensive contents for the interested readers.

We first summarize the changes related to improving the readability (Sect. 1). We then answer to each reviewers comment in Sect. 2.

We look forward to the feedback and thank you once again for the consideration of the manuscript.

Best regards,
Basil Kraft (on behalf of co-authors)

**Meta**

- We received reviews from four reviewers:

    - Lieke Melsen: Reviewer #1 (Sect. 2.1)
    - Anonymous: Reviewer #2 (Sect. 2.2)
    - Derek Karssenberg: Reviewer #3 (Sect. 2.3)
    - Anonymous: Reviewer #4 (Sect. 2.4)

- We applied minor reformatting to the comments to harmonize them, but without changing the content.

- We do not list minor comments (typos, etc.) pointed out by the reviewers, but they have been corrected.

- To avoid confusion, references within this response are highlighted in orange (e.g., Sect. 1.1 or Fig. 1).

- Whenever we refer to the old manuscript, we use the pattern *L. 122 [old]* or *Fig. 3 [old]*.

- When we reference the revised manuscript, we use the pattern L. 122 [new] or Fig. 3 [new].

**1 General changes**

**1.1 Complexity of the manuscript**

To reduce the complexity, we restructured the manuscript with the following general changes:

- Several figures were removed or simplified (see Sect. 1.2 for more details).

- The methods section was improved by adding a more detailed description of the neural network module (as requested by reviewer #1), and the training setup is described in more detail.

- The results section was shortened, mainly by removing and simplifying figures, as suggested by multiple reviewers.

- Some aspects of the discussion were merged and shortened.
    - The discussion on the hydrological responses was thinned out.
    - The subsection on *4.3.1 Uncertainties [old]* was removed, but the major points were integrated into a shortened section 4.3 Challenges and opportunities [new].

**1.2 Figures**

We identify with the suggestions that the manuscript was too detailed with complex figures. To improve these aspects, the following revisions were undertaken:

- *Figure 4 [old]* that showed latitudinal variations of phase and variance errors (shown here as Fig. 1a) was removed.

- *Figure 9 [old]* of the local response of evaporative fraction to soil moisture (shown here as Fig. 1b) was removed.

- *Figure 12 [old]* for the seasonality of regional TWS decomposition (shown here as Fig. 1c) was moved to Appendix B.

- *Figure 5 [old]* of regional TWS simulations and *Figure 6 [old]* of regional SWE simulations (shown here as Fig. 2a and Fig. 2c) were moved to Appendix B. The simplified versions, now Figure. 4 [new] (shown here as Fig. 2b) and Figure. 5 [new] (shown here as Fig. 2d).

- Fig. D1 [new] in Appendix D, showing H2M performance when using WFDEI forcings, was added (shown here as Fig. 3).

- In all applicable figures, axis labels were improved, the color of masked pixels in maps was changed, and legends were improved.

[Figure]

(a) *Fig. 4 [old]*          (b) *Fig. 9 [old]*          (c) *Fig. 12 [old]*

Figure 1: Former *Fig. 4 [old]* (a) and former *Fig. 9 [old]* (b) were removed. Former *Fig. 12 [old]* (c) was moved to Appendix B, now Fig. B3 [new].

[Figure]

(a) *Fig. 5 [old]*

(b) New Fig. 4 [new], replaces *Fig. 5 [old]*

(c) *Fig. 6 [old]*

(d) New Fig. 5 [new], replaces *Fig. 6 [old]*

Figure 2: Former *Fig. 5 [old]* (a) and *Fig. 6 [old]* (c) were moved to Appendix B, now Fig. B1 [new] and Fig. B2 [new], respectively. The figures were replaced by Fig. 4 [new] (b) and Fig. 5 [new] (d), the global average variables including the seasonality and the interannual variability.

[Figure]

Figure 3: The new Fig. D1 [new] in Appendix D shows the model performance of the H2M with the forcing variables used in the manuscript, and with the same ones (WFDEI) as used by eartH2O GHMs ("H2M (WFDEI)").

**2 Response to reviewer comments**

**2.1 Reviewer #1**

**General comments**
* * *
**(1.1) Complexity** *Unfortunately, the manuscript is quite hard to read, especially because in the figures several letters dropped of the axes, which made it a puzzle to find out what was shown where (I did not manage to solve this puzzle for Fig 10), this made it hard to estimate if all conclusions are robust / valid. Besides, some sections and choices are hard to follow for an average HESS reader with average ML knowledge (as I consider myself that way..). Below I indicate this in more detail, hopefully the authors will be able to improve and clarify this in a next version.*

**Reply**: Thank you for the detailed comments. We understand that the figures and presentation in the original manuscript may have been too complex, and also had upload errors. In the revision, we have put significant efforts to reduce the complexity. For general overview of the changes, please refer to Sect. 1.1. Figure changes (some were removed, some were simplified, general readability was improved) are summarized in Sect. 1.2
* * *
**(1.2) Terminology** *Expressing soil water as a deficit requires the reader to pay a lot of attention.*

**Reply**: We understand that the cumulative water deficit (CWD) versus soil moisture (SM) causes confusion. We help the reader by explaining the relationship in more detail in the section 2.2.1 Hydrological components / H2M storage components, L. 260 ff. [new]: "Negative CWD is loosely and conceptually interpreted as root zone soil moisture, as it serves as the moisture source for evapotranspiration. This is in fact consistent with the physical models, even though CWD does not have a continuous interaction with GW storage except during overflow in H2M. It should here be noted that soil moisture and CWD are interchangeably used throughout the manuscript.". In addition, we added "wet → dry" labels in Fig. 6&7 [new], added "SM (-CWD)" in Fig. 8&9 [new], and were more precise in the text (e.g., "Note that CWD represents a deficit and thus, it corresponds to negative soil water storage." in Tab. 2 [new] or "With an increasing soil water deficit (larger CWD, dryer soil), the soil recharge increases [. . . ]" in L. 480 f. [new].
* * *
**(1.3) Figures** *Please solve the axes issues for all figures. In Fig 4, for instance the N is missing on the y-axis, and the 40 and 60 have dropped off, and the legend is unclear (my guess is it should be H2M and GHMs). In Fig 5/6, first letters of the month dropped off x-axis, and it took a while before I realize the two middle panels show variation over the years (not only because the numbers dropped off, it would be helpful to add a label 'years', in the same way it would be helpful to add a y-axis label TWS or SWE). In Figure 10 I don't know which variable is on which corner of the pyramid. Perhaps updated figures can be uploaded in response to this review, so that other reviewers can use these figures.*

**Reply**: In the revision, all figures were further improved: we added axes labels, removed or simplified them (see Sect. 1.2), and changed the background color of masked regions in maps as requested by other reviewers.
* * *
**(1.4) Figure complexity** *Besides the axes-issues, I think the figures themselves are anyways challenging read. There is a very high information density in each figure, but the figures are often not directly showing what is most interesting.*

**Reply**: Figure changes (some were removed, some were simplified, general readability was improved) are summarized in Sect. 1.2.
* * *
**(1.5) Complexity Figure 5&6** *For Fig 5/6 for instance, one could consider to show the difference between the models and the observations in a barplot, rather than their temporal dynamics.*

**Reply**: We appreciate the suggestion. We believe that showing the dynamics is very important. We, therefore, decided to simplify by only presenting the globally averaged variables in the main text, while moving the regional figures to the Appendix (see Sect. 1.2) for more details.
* * *
**(1.6) Figure 2** *Figure 2 is only very very briefly introduced, even though the climate regions are extensively used for all figures.*

**Reply**: We agree that the regions are only briefly introduced. As length of the manuscript limits adding further text, we refer to the original publication (Papagiannopoulou et al., 2018), which is also cited in the manuscript, L. 385 [new].
* * *
**(1.7) Forcings** *It is unclear why the authors have decided to use the forcing from three different data sources, which makes the study more sensitive to inconsistencies / non-closure of balance, etc. Besides, it remains undiscussed how these data compare to the data used in the eartH2O project, because it might explain some of the differences with the GHMs.*

**Reply**: We agree that the uncertainties in the forcing data have a big impact on the model simulations. Owing to the exact same reason, we selected the forcing data that are as close as possible to observations, and different variables are unfortunately from different sources. To get a picture of how that would potentially affect the findings of our study, we retrained the H2M model with WFDEI (same as that used in eartH2O). As the results were very similar (see Fig. 3, or Fig. D1 [new] in the revised manuscript), we concluded that the findings are robust even with data from different sources.

**Detailed comments**
* * *
**(1.8) Masking** *Grid cells with large withdrawals have been removed but it is unclear which data source was used to identify cells with groundwater withdrawals.*

**Reply**: The masking is now described in L. 160 f. [new].
* * *
**(1.9) Static variables** *The procedure with the static input layers is unclear to me. First, they are compressed to 30 (l.95-100 on p4) and then from 30 to 12? (l.140 p7).*

**Reply**: We describe the procedure in more detail now in Sect. 2.2.2 The neural network (NN) module, L. 279–286. [new]. In short: the first reduction is a pre-processing step, while the further compression from 30 to 12 values is performed "online" as part of the model optimization.
* * *
**(1.10) Negative NSE** *In the validation, large negative NSE values were rescaled, but in Table 3 the spatial mean NSE is given. This makes the numbers provided here not comparable to NSE values obtained in other studies. Question is if it should still be called NSE, this can be misleading.*

**Reply**: We agree that the transformation of the NSE is not necessary. Instead of transforming the negative NSE values, we now truncate the y-axis of Fig. 3 [new] at -1, as the large negative values have no intepretable information. In *Table 3 [old]*, only one value was affected by the transformation, and more importantly, the findings and conclusions of the manuscript do not change.
* * *
**(1.11) Section 3.1** *In general, section 3.1 is hard to follow, because it is not directly clear what the spatially averaged signal is - is that averaged globally?*

**Reply**: We explain the calculation of the *spatially aggregated signal* in more detail in Sect. 2.4.2 Temporal and spatial scales, L. 391–402 [new]. The text is simplified as well.

**2.2 Reviewer #2**

**General comments**
* * *
**(2.1) Comparison to GHMs** *I am not sure that the comparison of H2M with GHMs is completely fair because the precipitation dataset used to force H2M (GPCP) is based on observations, while the one used to force GHMs is derived from the ERA-Interim reanalysis. Another reason why the comparison may be unfair is that spatial resolution of GHMs is degraded from 0.5 degree to 1 degree to be compared to the H2M simulations. Also, it should be emphasized that some GHMs are uncalibrated models.*

**Reply**: We appreciate your comments. This is a fair point. In the revision, we clarify that from the GHMs only LISFLOOD is calibrated using runoff at catchment level and that the comparison is not entirely consistent L. 446–450 [new]. we further highlight that the GHMs are used as a benchmark for H2M, and not to show which model(s) performs better. Our aim is validate H2M, and to show that it is

within the range of GHMs (with better local adaptivity, as expected), and that the TWS partitioning is qualitatively similar.
* * *
***(2.2) CWD vs SM*** *In the whole paper, there is a confusion between CWD and SM. E.g., L. 249 "We consider the dynamics of CWD to correspond to SM and thus, the terms are used interchangeably when talking about soil moisture dynamics": has to be clarified.*

**Reply**: We understand that the cumulative water deficit (CWD) versus soil moisture (SM) causes confusion. We help the reader by explaining the relationship in more detail in the section 2.2.1 Hydrological components / H2M storage components, L. 260 ff. [new]: "Negative CWD is loosely and conceptually interpreted as root zone soil moisture, as it serves as the moisture source for evapotranspiration. This is in fact consistent with the physical models, even though CWD does not have a continuous interaction with GW storage except during overflow in H2M. It should here be noted that soil moisture and CWD are interchangeably used throughout the manuscript.". In addition, we added "wet → dry" labels in Fig. 6&7 [new], added "SM (-CWD)" in Fig. 8&9 [new], and were more precise in the text (e.g., "Note that CWD represents a deficit and thus, it corresponds to negative soil water storage." in Tab. 2 [new] or "With an increasing soil water deficit (larger CWD, dryer soil), the soil recharge increases [...]" in L. 480 f. [new].

**Detailed comments**
* * *
***(2.3) H2M (Abstract)*** *Is H2M a new model developed in this study? What is the added value of this approach with respect to more traditional modeling approaches? What is the meaning of H2M acronym?*

**Reply**: We improved the abstract and included the points mentioned. E.g., "This combination of machine learning method and physical knowledge can potentially lead to data-driven, yet physically consistent and partially interpretable hybrid models" in L. 9–10 [new] and "The hybrid hydrological model (H2M), extended from (Kraft et al., 2020) [...]" in L. 11 [new].
* * *
***(2.4) L. 95 (22 static variables)*** *unclear because 4 lines correspond to static variables in Table 1, not 22.*

**Reply**: This is now stated more clearly in Sect. 2.1.2 Static variables, L. 101 [new] ("(6 variables in total)" etc.): the variable groups contain multiple variables, e.g., different soil properties.
* * *
***(2.5) L. 166, 174 (soltmax, softplus)*** *all readers may not be familiar with these machine learning technical terms. They should be defined.*

**Reply**: They are now defined, softmax in L. 206 [new], softplus in Eq. 3, L. 191 [new].
* * *
***(2.6) L. 199 (model training)*** *more details should be given on the used machine learning approach. Is a local training (one statistical model for each model grid cell) performed or a global training (all model grid cells together represented by the same statistical model)?*

**Reply**: The section on model training, Sect. 2.3 Model training [new], was improved and should be clearer now. The H2M is global, i.e., the model learns global water cycle dynamics, which is clearly stated in Sect. 2.2 The hybrid hydrological model (H2M), L. 167 f. [new] of the revised manuscript.
* * *
***(2.7) L. 243 (Table 2)*** *CWD and SStor are written here for the first time and were not defined before. A clear definition should be given. The definition of CWD given in the next paragraph is not clear.*

**Reply**: The definitions of CWD and SStor are now given in the caption of Tab. 2, and in L. 148, [new] (SStor) and L. 260 ff. [new] (CWD).
* * *
***(2.8) L. 242 (selection of models)*** *How was model selection made? In Schellekens et al., 10 models are considered.*

**Reply**: We only selected the models for which groundwater storage was available, which is now clarified in L. 143 ff. [new].
* * *
**(2.9) L. 284 (Table 3)** *the period of time for which the comparison was made should be indicated.*

**Reply**: The time period (2009 to 2014) is now mentioned in the table caption. Also, the differences in time periods are also clarified in wherever applicable at the beginning of sections.
* * *
**(2.10) L. 286 (model intercomparison)** *Could be completed with a water balance Table similar to Tables 7 and 8 in Schellekens et al.*

**Reply**: A water balance table was added, Tab. 4 [new].
* * *
**(2.11) L. 293** *Mean or median scores are little informative in case of non-Gaussian score value statistical distribution. Could you plot score histograms instead?*

**Reply**: The evaluation of the globally averaged variable is commonly used to assess the models ability to represent the integrated signal. We added the following statement in the revised version: "The global scale signal are themselves useful indicators, as they are often used to characterize the Earth system and land surface processes, e.g., climatic changes (Pachauri et al., 2014), or to evaluate water-carbon relations (Jung et al., 2017; Humphrey et al., 2016)." (L.396 ff. [new]). The median shows how well the model performs on the local level, although, as you say, the variability matters. We show the spatial variation of the cell-level performance in Fig 3 [new], but adding similar figures for all the metrics would make the manuscript and figures even more complex. As several reviewers suggested the opposite, we have simplified things. Unfortunately, We therefore decided not to add the quantiles to the text. This is also in line with our main focus of interpreting the responses of hydrological variables using latent variables from machine learning, and benchmarking them against established GHMs.
* * *
**(2.12) L. 340 (Amazon basin)** *the Amazon area was affected by droughts (2005, 2010, 2015). Are these drought events visible in the simulations performed in this study?*

**Reply**: Fig. B1 [new] shows the Amazon region (T1 S-AM tropical) in detail. From the years you mentioned, only 2005 and 2010 are covered by our simulations. In both cases, the H2M model, in fact, reproduces the GRACE patterns well.
* * *
**(2.13) Figure 10** *CWD is indicated as one of the 3 considered variables, while the Figure itself shows SM.*

**Reply**: We have now added a clearer label "SM (-CWD)" to *Fig. 10 [old]* (now Fig. 8 [new] in the revised version).

**2.3 Reviewer #3**

**General comments**
* * *
**(3.1) Complexity** *[The manuscript] needs considerable revision in particular regarding the presentation of the material: figures are often very unclear (it is sometimes even unclear what attributes are shown), the text is rather long and could be condensed providing at the same time more focus. Regarding the latter: this seems to be a proof-of-concept paper. It is thus not essential to provide a complete evaluation of the model. Instead, I believe it is more important to properly explain the methodology and key outcomes. In revising the paper I suggest the authors to possibly leave some of the results out (e.g. less figures, simpler figures) – it wouldn't harm the paper but may make it more accessible.*

**Reply**: We would like to thank the reviewer for the comments and suggestions. In the revision, we have considered all the suggestions made here. In particular, methodological aspects are made clearer; results sections are simplified to highlight the main points with some contents removed and/or moved to the appendix; and the discussions condensed to a more digestible form. See Sect. 1 for more details.
* * *
**(3.2) Figures and tables** *Most of the figures are hard to understand. Legends are often missing, panels are included without any explanation of what is shown, variables are plotted for which it has not been explained how they are calculated. This needs considerable improvement. Please enlarge the figures as well. In my detailed comments I pin-point a few things that are not clear, please consider this as examples (not a complete list).*

**Reply**: We agree that some figures had too much information, but the notations and explanations were lacking. From the comments, we could not pin-point the exact figure with missing legends. Nevertheless, we have revised and improved all figures, as also suggested by other reviewers. In general, we added axes labels where they were missing, removed or simplified figures in general (see Sect. 1.2), improved visibility of the legends, etc.
* * *
**(3.3) Objective function** *The objective function contains four different observational data (terrestrial water storage, evapotranspiration, runoff, snow water equivalent). It is completely unclear how each of these are weighted in the objective function. Do you 'calibrate' against standardized values of these attributes? If so, how are these standardized? Please explain. Note that it is to be expected that this weighting has a strong influence on the results, e.g. if more 'weight' is assigned in the objective function to runoff, the model will perform better in runoff prediction. I do not expect you to explore different objective functions but at least the objective function needs to be given and it needs to be explained that this is quite arbitrarily chosen. Note that for instance in Bayesian data assimilation 'weights' of observations will depend on the uncertainty associated with the observations (high uncertainty -> low weight). This is not the case in your approach.*

**Reply**: As mentioned in the manuscript, we used uncertainty-weighted task weighting. In the revised manuscript, we have explained this aspect in more detail in Sect. 2.3.3 Multi-task loss [new].
* * *
**(3.4) Training, validation, testing data sets** *In machine learning, one uses training and validation sets in the procedural step of model building. Model evaluation then is done using a data set not used for model building (this is often referred to as the test data set in the machine learning world). It seems you are not separating validation and testing data sets. This is an important issue – evaluation of the model (all the performance metrics provided, almost all plots provided with model outcomes) needs to be done on data that are not at all involved in the model building phase. If you are evaluating on the same data as used for building the models, this should be clearly indicated in the manuscript and implications of this extensively discussed.*

**Reply**: Originally, we referred to Kraft (2020), where the spatio-temporal splitting into training, validation, and test set is explained in detail. In the revision, we have also outlined the approach in this manuscript Sect. 2.3.1 Cross-validation setup [new] to make it complete.
* * *
**(3.5) Context, aim of modelling** *I could argue that in your comparison of your approach to modelling (hybrid modelling) and existing global hydrological models you are comparing apples and oranges. The main aim of hybrid modelling (as defined in your manuscript) is prediction (in the statistical sense): estimating variables at time steps for which observational data are not available. Existing global hydrological models however aim at scenario analysis (and oftentimes prediction as well). Scenario analysis may involve evaluating effects of climate change, effects of future changes in water allocation (e.g., irrigation, domestic water use), effects of future changes in land use, etc. Hybrid modelling is not suitable at all for scenario analysis as it almost completely relies on observational data on the system. If the system changes, it won't work anymore. I may exaggerate somewhat (to make my point clear), and you may disagree in which case I challenge you to convince me otherwise. In any case I suggest you 1) mention this difference in the introduction of the manuscript and 2) discuss this issue in the Discussion section. I consider this important in particular because this is a 'proof-of-concept' paper and it is thus important to position this work in the broader context.*

**Reply**: Perhaps, we may not have been clear enough in the manuscript, but the comparison of H2M with GHMs was actually intended as a validation of the H2M itself rather than as an effort to provide an alternative GHM. The holy grail of hybrid modeling would be to bring tightly together the best of machine learning methods and the physical models. For example, the mass balance and hydrological responses in machine learning methods, while still having a robust data adaptability that are not always the case in physical models with process simplification and abstraction and rigidity of physical/empirical parameters. Future predictions are a challenge in general even for physical models, as seen through numerous generations of model intercomparison projects. Even the validity of the "physical" parameters (that are often learnt through iterations of model development) can be an interesting puzzle to solve. But, given that for GHMs, we still cannot state that machine learning methods are better. In fact, the data-space that the machine learning methods are trained with may provide a range of climate-spectrum that may be within the projected changes in climate. A particular aspect of the "interpolation" vs "extrapolation" by

machine learning methods is discussed well in Jung et al. (2020, `doi.org/10.5194/bg-17-1343-2020`). Lastly, one could also argue that the data-adaptivity of the machine learning methods give them a larger flexibility in dealing with non-linear responses (as touched upon in the hydrological responses section of the manuscript) of the land surface to changes in climate. We, however, confide that more research on the applicability of the hybrid models on scenario-type analysis is necessary. Only such research would be able to make a convincing argument rather than a speculative one we can make now based on the state-of-the-art. We hope that our proof-of-concept can contribute towards that. We added a statement that clearly states the aim of this study in L. 68–78 [new], and we believe that the critical in-depth evaluation of the learned parameters and hydrological responses underlines our goal to, in the future, use the hybrid approach to achieve data-driven insights into environmental processes. At several instances, we have also clarified that the comparison with GHMs is a benchmarking of H2M rather than the direct comparison of the performance metrics.

**Detailed comments**
* * *
**(3.6) End of introduction** *The introduction gives a good overview of past work in the domain. However, on line 68-72 it remains somewhat unclear what the contribution of this paper is. I suggest stating this more explicitly and also to state more explicitly what this paper adds compared to your previous publication (Kraft et al., 2020).*

**Reply**: Thanks for the suggestion. We have added a short paragraph at the end of the introduction to clarify the merit and goal of the study in L. 68–78 [new].
* * *
**(3.7) Code sharing** *I strongly suggest sharing the code of your model on a public repository (e.g., GitHub). It will make your work more credible and will enable other researchers to build on your research.*

**Reply**: (The code and simulations are available online (see Code and data availability statement, L. 720 [new]) at `github.com/bask0/h2m`.)
* * *
**(3.8) L. 113** *Clearly state what Q refers to. It is the amount of runoff generated in a pixel (or area of land), not the discharge from the pixel (streamflow). The latter can only be calculated in a spatial model that does channel routing.*

**Reply**: In the revised manuscript, we added the following statement to describe Q in more detailed: "Note that only catchments with an area similar to the spatial resolution of the meteorological forcings were used for the prediction and thus, Q does not include larger routed streamflows and only provides an estimate of gridded runoff." (L. 132–134 [new]).
* * *
**(3.9) L. 123** *Adjust the numbering, is this a nested numbered list?*

**Reply**: The paragraph was rephrased, L. 156–164 [new].
* * *
**(3.10) Model description** *Is this a spatial model, i.e. does it include spatial interactions in the time transition functions? I don't think so, it is a local (point model). Please state this clearly.*

**Reply**: The model is a global one but there are no spatial interactions (point model). This is now stated more clearly in the manuscript, L. 167–170 [new].
* * *
**(3.11) Figure 1** *The caption is too long. Reduce it and explain concepts in the main text.*

**Reply**: The caption has been shortened. Note that the listing of the variables could not be reduced.
* * *
**(3.12) Neural network** *I am unsure you are building the machine learning model for all cells at once (single model) or for each cell separately (number of models equals number of cells). If the former, the method you propose is, I believe, fully non-spatial (point model for hydrology, identified separately for each cell with observations for that cell). Please explain this clearly.*

**Reply**: The section on model training, Sect. 2.3 Model training [new], was improved and should be clearer now. The H2M is global, i.e., the model learns global water cycle dynamics, which is clearly stated in Sect. 2.2 The hybrid hydrological model (H2M), L. 167 f. [new] of the revised manuscript.
* * *
**(3.13) L. 196** *Why DELTA T instead of T?*

**Reply**: Our notation was a bit imprecise. We compare the $T$ dynamics, but not the mean, and thus we could say that we model $\Delta T$ in time. But, in fact, what we do is: remove the mean from T and S + G + (-C) independently, such that only the modeled dynamics are compared (because T is not absolute terrestrial water storage but just the variations). We improved the notation in the revision (Fig. 1 and L. 252–257 [new].
* * *
**(3.14) Model training** *I am wondering how you train the model. Machine learning models typically do not run forward in time. However, in this application, they are fed by temporally changing data, in a forward timestep approach. How is this done? Please explain. Providing the code would help as well.*

**Reply**: Model training is now described in Sect. 2.3.2 Training setup [new]. The code is also available and the link is provided in the manuscript. We additionally show the pseudo-code in Appendix E [new] to illustrate the model training.
* * *
**(3.15) Figure 2** *Too small.*

**Reply**: We have increased the size of Fig. 2.
* * *
**(3.16) Section 2.5** *Which runs of the global hydrological models were used?*

**Reply**: We use the version WWR1 that are available at `https://wci.earth2observe.eu/`
* * *
**(3.17) Caption Table 3** *What is 'median-cell level'? Please explain.*

**Reply**: It is the median across all grid cells, i.e., metrics are calculated per grid-cell first, and the median is reported. The spatially averaged signal is the global area-weighted average (i.e., mean across all grid cells, from which the metrics are calculated). This is now clearly explained in Eq. 27 and 28 [new].
* * *
**(3.18) Figure 3** *Why are you not including results for ET and Q as well?*

**Reply**: ET and Q are not direct observations, and are up-scaled from local measurements, and can be classified as observation-based products, but on the basis of ML methods. We based our original decision on the assumption that performance for these variables is more prone to be affected by upscaling prediction uncertainties than for those variables obtained more directly and independently from remote sensing. We provide ET and Q metrics in Tab. 3 [new], and show the local biases in Appendix A, Fig. A2 [new].
* * *
**(3.19) Figure 4** *Explain root phase and variance error in Methods. I am in the opinion however this plot could be left out.*

**Reply**: We agree and have removed Fig. 4 from the revised manuscript.
* * *
**(3.20) Figure 5** *What are the small panels on the left and right side? What is plotted on the x-axis? The figure is hard to understand. It needs to be simplified (and possibly include more detailed information in a digital supplement or appendix).*

**Reply**: Fig. 5&6 [old] were simplified (now Fig. 5&6 [new]), see Sect. 1.2 for more details. The panels ("inset axes") show the NSE per model (color corresponds to model). If you were referring to the outer panels, they were for the mean seasonal cycle.
* * *
**(3.21) Figure 7** *What is represented by each line in the figure? A single location? A single year? 'quantile of the spatio-temporal distribution' is not easy to understand.*

**Reply**: We "mix" all time steps and grid cells and calculate the quantiles as bins of CWD. For example, we filter all the values (space and time) for CWD values from 0 to 10 mm. From these values, we calculate

the quantiles (independently per cross validation run). Then we take CWD from 10 to 20 mm, etc. The caption is improved in the revision (now Fig. 6 [new]).
* * *
**(3.22) Figure 8** *What is represented by the colours?*

**Reply**: As the legend indicates, the lines represent dry (red) and wet (green) conditions. The Figure (now Fig. 7 [new]) has an improved legend and the colors are explained in the caption.
* * *
**(3.23) Figure 10** *Too small. Consider selecting runs and plotting these.*

**Reply**: We increased the figure size (now Fig. 8 [new]). Unfortunately, We could not understand the suggestion on selecting runs.
* * *
**(3.24) Figure 11** *Too small. What is along the x-axis? Months? It runs up to 13.*

**Reply**: We increased the figure size (now Fig. 9 [new]), in which the axes are properly labeled.
* * *
**(3.25) Discussion section** *The discussion is interesting but it is somewhat long. Consider reducing it somewhat focusing on the main things (that are relevant to the research objective and questions).*

**Reply**: We agree. We have now shortened the discussion, please see Sect. 1 for a summary of changes.

**2.4   Reviewer #4**

**General comments**
* * *
**(4.1) Complexity of results and figures** *The contribution is novel and fits well within the scope of HESS, however requires major revision before potential publication. This especially concerns the results section which contains a large number of (complex) figures that are often difficult to read and/or grasp. The reader is required to go back and forth between the main text and the caption in order to understand what is displayed. These comments are already based on the revised set of figures uploaded by the authors. I would recommend to thoroughly revise the results section, potentially drop a few figures and revise the remaining ones, in order to arrive at a more concise and digestable presentation.*

**Reply**: Thank you for an overall positive feedback on the study. We have incorporated the suggestions, and revised the manuscript significantly, mostly to clarify and simplify the results. The figures have been moved, removed, replaced, re-drawn (see Sect. 1.2). Consequently, the text have also been streamlined and, we believe, the manuscript is much more concise now.
* * *
**(4.2) Complexity of discussion** *The discussion touches many important aspects, however appears lengthy and repetitive at times. I would recommend revision to make it more concise.*

**Reply**: We shortened the discussion and merged its subsections, please see Sect. 1.1 for more details.
* * *
**(4.3) Parameter estimates** *The significance of the parameter estimates for process-based modeling is overstated in my opinion. The authors acknowledge that, due to the simple model structure and small number of parameters, parameter estimates will tend to compensate for insufficient or lacking process representations and uncertainty in the input data, which undermines their 'physical meaningfulness' and ability to describe specific processes.*

**Reply**: The interpretability of the parameter estimates is the major focus of this study. By comparing them to prior knowledge (Sec. 4.2.1 Hydrological responses [new]), we show that the parameters and latent responses are indeed interpretable to a certain extent. This is discussed extensively. We agree that some skepticism is justified, and we carefully evaluate the current issues with the approach. We think that this study constitutes a first step towards data-driven retrieval of parameters. Given that this should still be viewed as a comprehensive proof-of-concept, we do not fully agree with this comment. However, we agree that the model needs to be improved as discussed. Regarding the relevance of parameters in process-based models, we wanted to highlight that the model parameters are a critical source of uncertainty as well (e.g., Liu and Gupta, 2007, doi: 10.1029/2006WR005756). Several related specific instances are mentioned throughout the text for covering different aspects of "parameter uncertainty," such as equifinality issues,

scale mismatch between the estimation of parameters (such as hydraulic conductivity in the lab vs the same at half degree resolution), discrete vs continuous in space (e.g., vegetation type and a corresponding lookup table based on observations at plant/ecosystem level), and temporally fixed vs varying. In general, we argue that hybrid modeling can, in the future, provide model parameters, but not in the current state without further research 653–655 [new].

**Detailed comments**
* * *
**(4.4) Masking anthropogenic impact** *It remains unclear which thresholds and data-sets were used to identify cells with "high anthropogenic impact". Groundwater abstraction is given as an example, however cells with extensive irrigation (irrespective of source) should be removed due to its effects on soil moisture and evapotranspiration.*

**Reply**: We used an ad-hoc solution based on Rodell (2019, nature.com/articles/s41586-018-0123-1). We removed all regions with only groundwater depletion due to anthropogenic activities (#7, #12, and #14). We added this information to the revised manuscript, L. 160 [new].
* * *
**(4.5) Global parameters** *It remains unclear if the parameters $\beta_s$ and $\beta_g$ were estimated by the neural network or were preset. In the latter case, please clarify how these parameters were determined.*

**Reply**: They are estimated as free parameters (in the sense of not being connected to data), i.e., they are not predicted by a neural network but rather by the optimizer. The gradient-descent-based optimizer receives all parameters of the neural networks (NN) and the global parameters. In each optimization iteration, the optimizer can update the weights of the neural network, as well as the two free parameters. We have added more details to the manuscript, in 263–366 [new].
* * *
**(4.6) Global metrics** *It is my understanding that global, area-weighted averages of TWS, SWE, Q, and ET have been used to calculate the performances reported in Tab. 3 and Fig. 3. If correct, I cannot quite see the value in doing so. Both H2M and the GHMs aim to estimate hydrological states and fluxes in a spatially distributed manner, i.e. numbers based on a global average provide little insight into the models' performances. Further, jumping between global performance and cell median makes Sect. 3.1 rather hard to follow. I'd suggest to focus on cell median and to ditch the global numbers.*

**Reply**: We prefer to keep them in the text, but we added a more detailed explanation of the calculation and a justification to use global and local metrics (391–402 [new]).
* * *
**(4.7) Evaluation on training set** *The comparison of model performances between H2M and the GHMs in Fig. 3 seems little meaningful since the better part of the common time series (2003-2008) was part of the training data-set; particularly since NSE and MSE are closely related. In this regard, it would also be interesting to see a direct comparison of the performances achieved by H2M in the training period and the evaluation period, respectively.*

**Reply**: The quantitative performance is calculated on the test set (2009-2014), or on the common time range (2009-2012) when comparing with the GHMs. For the qualitative assessment, Fig. 4, 5, and 9 [new], and the TWS decomposition maps, Fig. 8 [new], the training time range was included. This is justified by the minimal generalization error of H2M from the training to the test set. Also, longer time ranges give a more robust picture. We have added the information of time period to each results subsection in the revised manuscript.
* * *
**(4.8) Figure 3** *Fig. 3 is hardly readable, please rescale/revise.*

**Reply**: We increased the size of Fig. 3 [new].
* * *
**(4.9) Figure 4** *Fig. 4 shows performance metrics that have not been introduced in the methods section or used in the previous figures and tables which, frankly, is confusing. I'd recommend to stick with the performance metrics used earlier.*

**Reply**: This figure has been removed.
* * *
**(4.10) Figure 5&6** *The insets in Figs. 5 and 6 severely compromise readability and I'd suggest removing them. Further, the x-axis labels seem to be cut off. Please revise.*

**Reply**: Fig. 5&6 [old] were simplified (now Fig. 4&5 [new], please see Sect. 1.2 for more details; The original figures were moved to Appendix B with improved axis labels.
* * *
**(4.11) Figure 7** *Fig. 7: Q is an unfortunate abbreviation for quantile here since used for runoff in other parts of the manuscript. Please revise. Which variable/quantity are the quantiles exactly based on? Please clarify.*

**Reply**: We now use "percentile" in the Fig. 6&7 [new] (former Fig 7&8 [old]). Also, the captions were improved to describe the calculation of the percentiles in more detail.
* * *
**(4.12) Figure 9** *Fig. 9: The masking color (black) and the darkest shade of the color scale are hardly distinguishable, please revise. In general, I feel that the figure conveys a similar message as Fig. 8. Given the overall large number of figures in the manuscript, this one could be dropped for conciseness.*

**Reply**: We changed the masking colors in the maps in Fig. 2 [new] and Fig. 8 [new] (former Fig. 10 [old]) and removed Fig. 9 [old].
* * *
**(4.13) Figure 10** *The masking color (grey) is also part of the color scale (equal contribution from all three components), please revise.*

**Reply**: We changed the masking color in Fig. 8 [new] (former Fig. 10 [old]) to white.
* * *
**(4.14) L. 89** *Please rephrase "average content [···] of bulk density".*

**Reply**: Rephrased to [...] and the average (along depth) of [...] (L. 100 [new])
* * *
**(4.15) L. 95** *Please rephrase "keep".*

**Reply**: Rephrased to "These 20 static variables were spatially aggregated from their finer resolution to 1/30∘ to maintain sub-grid variations, yielding a block of 30 latitude cells times 30 longitude cells times 20 variables, [...]" (L. 107 ff. [new]).
* * *
**(4.16) L. 127** *What percentage of global land area do the remaining 12 084 grid cells represent?*

**Reply**: About 80%, which is now mentioned in L. 164 [new].

---

## Author Response (AR2)

**Minor revisions for the submission**
**hess-2021-211**
**"Towards hybrid modeling of the global hydrological cycle"**

**Kraft et al.**

**January 25, 2022**

Dear Editor, dear Reviewers,

Your comments and suggestions helped us tremendously to improve the manuscript. We highly appreciate your time and efforts. Also, we are glad that you acknowledge our efforts to improve the study.

In this second review, we received comments from three referees:

- Anonymous #1: Suggested *accepted as is* with no comments.

- Derek Karssenberg: Response in Section 1.

- Anonymous #2: *Comments via email.*: Response in Section 2.

Please find our response to the comments below. In addition, several typos were fixed and minor changes were applied to improve text flow.

Kind regards,
Basil Kraft (on behalf of co-authors)

**1 Response #1**

**General comments**
* * *
***(1.1) General comment*** *The manuscript has been thoroughly revised and the authors have dealt with most of my comments (except one, see below) in a satisfactory manner. In my opinion this is an important contribution as it provides a proof of concept for the approach and a roadmap for future research. Please find below my comments related to the revised manuscript:*

**Reply**: We are very grateful for your comments. Your previous and current suggestions helped us a lot to improve the manuscript.
* * *
***(1.2) Context, aim of modelling*** *Related to my comment (and your response) on the original version of the manuscript, item 3.5 'Context, aim of modelling' I agree with your response to my comment. However, in my opinion a short (!) paragraph on this could be added to the discussion, in particular on how informative or valuable 'hybrid models' are expected to be compared to 'process-based models' (GHM). You could discuss the use case when models are used for prediction (or reconstruction) alone (state estimation), when models are used for improving our understanding of mechanisms, and when models are used for scenario analysis. Regarding the latter, I would like to note that scenario analysis does not only involve climate change scenarios (as referred to in your rebuttal), it also includes scenarios that may require a change in landcover/land use (e.g. due to biofuel expansion), a change in allocation of water over industrial, urban, agricultural water use, and possibly other policies or water management scenarios.*

**Reply**: We added the following paragraph to the discussion:
L. 680ff Global hydrological models are often used for different tasks such as the assessment of the water cycle at past and present, predictions for the future, for evaluating implications of, e.g., land use changes

by scenarios, and to gain process-understanding. In principle and technically, a global hybrid hydrological model can be applied for the same tasks while related simulations need to be interpreted with care. The strongest use case of H2M is the assessment of recent variations of the water cycle since it can act as a physically consistent yet data-adaptive bridge between heterogeneous global data streams and complements traditional data assimilation approaches. Interpreting predictions too far into the past or future can be risky when factors that are not represented physically play a role that had little impact during the training period (e.g., permafrost melting, $CO_2$ fertilization of water use efficiency). Likewise, scenarios of, for example, different land use could make sense to conduct if the conditions represented by the scenarios have been represented during learning in some way while there always remains the danger that learned relationships by the neural network are just statistical associations rather than causal relationships ("shortcut learning", Geirhos et al., 2020). As we could show, gaining process understanding from the hybrid model can be feasible as the spatially and temporally varying coefficients learned by the neural network are plausible and partly very interesting. However, such uncovered patterns may rather represent hypotheses that should be tested with complementary approaches like physical process modeling, direct observations, or experiments.
* * *
**(1.3) Abstract, line 14** *Runoff (Q) -> runoff generation (Q). In my opinion it is important to be very clear on the fact that streamflow discharge of large rivers is not modelled, neither is it used for constraining model parameters or process representations. I suggest changing this in Table 1 as well.*

**Reply**: We changed "Runoff (Q)" to "Grid cell runoff (Q)" in the abstract and in Table 1 to emphasize that it does not include discharge. We are hesitant to use the term "runoff generation (Q)" because it refers to a set of processes.
* * *
**(1.4) Runtime** *Runtime is quite a down to earth matter but it in practice it is important. Many GHMs have long runtimes and this is one of the reasons they are not extensively calibrated. What is the typical runtime (and hardware requirements) of a training (and prediction run without training) run for H2M? I suggest adding this somewhere in the manuscript (e.g. in the Results section and possibly in the Discussion section).*

**Reply**: We agree that this is a relevant fact and we added it to the manuscript:
L. 422ff An optimization run of a single cross-validation iteration takes about 6 hours, a forward run for all grid-cells and the entire period from 2002 to 2014 takes about 15 minutes. Each model was run on a NVIDIA Tesla Volta V100 16 GB GPU with up to 10 CPUs (Intel(R) Xeon(R) CPU E5-2698 v4 @ 2.20GHz) for data buffering and background tasks.
* * *
**(1.5) Figure 1** *In my opinion the figure does not clearly explain the general concepts, because it gives too many details (in the c) panel). Consider leaving out the equations and instead giving only parameter (and key variable) names. The figure should show the links between the hydrological model (in particular its parameters as these are linked to the neural net) and the neural network (and input data), not more. Equations are in the text so they do not need to be included in the Figure, in my opinion. Note that the figure gives $\alpha_{snow}$, I think this should be $\alpha_{smelt}$ (equation 2).*

**Reply**: We simplified Figure 1 as suggested and changed $\alpha_{snow}$ to $\alpha_{smelt}$. To indicate the usage of the coefficients and global parameters in the balance equations, they were added in parantheses. We updated the figure caption accordingly.
* * *
**(1.6) p. 9, line 189** *'the neural network'. There is more than one. Same on line 205.*

**Reply**: The "neural network" refers to the entire module, we changed it to "neural network module".
* * *
**(1.7) p. 12, line 278** *I suggest leaving out the URL here (www.deeplearningbook.org).*

**Reply**: The URL was removed as you suggested.
* * *
**(1.8) p. 13, line 314** *The cross-validation setup implies that validation sets are in the spatial domain very close to tuning data sets; offset of 1 pixel. As pixels close to each other will be similar, it is not so surprising (arguably) the validation works well (and thus possibly overfitting); training and validation data sets are not completely independent, they represent similar combinations of states, fluxes and parameter*

*values. An alternative would be training for instance in continent A (all pixels) and validating in the other continents. Please consider discussing this if you agree this may have an effect on the results.*

**Reply**: We agree with your statement. However, the discussion is already quite lengthy and the topic of cross-validation with spatio-temporal datasets is a complex one. Of course, we do not reduce the dependeny completey, but we tried to chose a "fair" setup for the comparison with the GHMs. From different experiments, we got the impression that the physical constraints and the multi-task training exert a strong regularization on the model, which reduces the data-adaptivity and thus overfitting. This is, however, just anectodal and needs further investigation. We added the following statement to the manuscript:
L. 327f Note that the spatial splitting reduces the dependency between the cross-validation sets, but does not completely remove it.
* * *
**(1.9) p. 13, line 322** *Validation runs. You may want to emphasize somewhere that the approach also enables time varying parameters when no tuning is done. This is because the neural network does a state update just like any other forward simulation model. I am sure you find this obvious, but it may not be that obvious for some readers*

**Reply**: We added the following sentence:
L. 174f For inference (after the optimization of the neural network), the model can be applied to unseen data like any forward simulation model without further model tuning.
* * *
**(1.10) Figure 6** *This is a very interesting figure of course. It shows how model parameters change with soil water deficit. This is extensively described in the text. However, it also shows considerable variation between pixels (or time steps) with the same soil water deficit (e.g. completely wet soils may have an $\alpha_{soil}$ between 0.2 and 0.8). I am wondering whether this variation entails mainly spatial variation or mainly temporal variation (or both). Please consider adding this to the manuscript (without extending it too much).*

**Reply**: We added the following statement to the manuscript:
L 489ff The relatively large variation under wet conditions (low CWD) in Fig. 6 can be attributed about equally to temporal and spatial variability. The groundwater recharge fraction $\alpha_{gw}$ shows a slightly larger temporal variability than the other fractions, and the contribution of the temporal component was generally a bit lower in the transitional regions.
* * *
**(1.11) Figures** *In general, the figures are still quite small, please be sure they appear in the right size in the final manuscript.*

**Reply**: We revisited every figure and improved readability. The latex template wants us to use 8.3cm for one-column and 12cm for full width figures, which is only a fraction of the full width, and we tried our best to follow these suggestions. It will not be an issue in the online version, and we think the figures are still readable in the print version.
* * *
**(1.12) p. 23, line 506** *PC-GLOBWB → PCR-GLOBWB*

**Reply**: This typo was fixed.
* * *
**(1.13) p. 27, line 585** *indicates → indicate*

**Reply**: This typo was fixed.
* * *
**(1.14) p. 41** *Consider acknowledging the reviewers.*

**Reply**: Thanks for pointing this out. Of course, we should acknowledge the efforts taken by the reviewers and the editor.

**2 Response #2**
* * *
***(2.1) General comment*** *The authors have done an excellent job at revising their initial submission. The manuscript has been restructured and the figures have been improved in way that makes it concise and easy to follow (which was the largest shortcoming of the initial submission as noted by all referees). All of my comments on the initial submission have been addressed in the rebuttal letter and in the revised manuscript and I recommend publication in HESS.*

**Reply**: Thank you very much for taking the time to help us improve the manuscript and for appreciating our efforts.
* * *
***(2.2) Eq. 21*** $L_v$ *should read* $L_C$?

**Reply**: The lefthand side of the equation was $\mathcal{L}_{v=C}(f_{\phi,\beta}, \boldsymbol{x})$, with $v = C$ in the subscript. To avoid confusion, we changed the equation (now Eq. 20) as you suggested.
* * *
***(2.3) L. 489*** *refers to arid and semiarid climates in Fig. 7, however the classification distinguishes boreal, temperate, transitional and tropical. Please clarify.*

**Reply**: We agree and changed the sentence. It now reads:
L. 495f In arid (S1-2) and semiarid (N1-5) climates, $\alpha_{et}$ exhibits a large range with steep gradients given low water input ($w_{in} = 0$ mm), decreasing with larger CWD (dryer soil).
* * *
***(2.4) L. 526*** *Which data biases are meant here?*

**Reply**: We added a paragraph on data biases and uncertainties:
L. 545ff In the hybrid modeling framework, the quality of the observational constraints is a major source of uncertainty. The data used in this study have well-documented deficiencies: The precipitation product, for example, shows large uncertainties in Africa due to limitations in density and quality of measurement sites (Sylla et al., 2013) and exhibits biases in snowfall estimates in the Northern Hemisphere due to over-correction of snowfall under catch (Behrangi et al., 2016; Panahi and Behrangi, 2019). The GlobSnow SWE saturates above 120 mm and underestimates the interannual variability (Luojus et al., 2010). TWS quality is generally difficult to quantify as an equivalent ground-based measurement does not exist, and its complex preprocessing has known impacts on the data quality (Scanlon et al., 2016). The machine learning–based constraints of Q and ET are not directly observed and thus, they are expected to have considerable global and regional uncertainties and biases (Ghiggi et al., 2019; Jung et al., 2020). This could lead to inconsistencies in the water balance (Trautmann et al., 2021). However, the multi-objective optimization may dampen negative effects of biases, as the model can trade off the different constraints.